# Novel charged sodium and calcium channel inhibitor active against neurogenic inflammation

**Seungkyu Lee[1†], Sooyeon Jo[2†], Sébastien Talbot[3], Han-Xiong Bear Zhang[2], Masakazu Kotoda[1], Nick A Andrews[1], Michelino Puopolo[4], Pin W Liu[2], Thomas Jacquemont[1], Maud Pascal[1], Laurel M Heckman[1], Aakanksha Jain[1], Jinbo Lee[5], Clifford J Woolf[1,2]\*, Bruce P Bean[2]\***

[1]FM Kirby Neurobiology Research Center, Boston Children's Hospital, Boston, United States; [2]Department of Neurobiology, Harvard Medical School, Boston, United States; [3]Département de Pharmacologie et Physiologie, Université de Montréal, Montréal, Canada; [4]Department of Anesthesiology, Renaissance School of Medicine at Stony Brook University, Stony Brook, United States; [5]Sage Partner International, Andover, United States

**\*For correspondence:**
clifford.woolf@childrens.harvard.
edu (CJW);
bruce_bean@hms.harvard.edu
(BPB)

[†]These authors contributed
equally to this work

**Competing interest:** See
page 23

**Reviewing editor:** Kenton J
Swartz, National Institute of
Neurological Disorders and
Stroke, National Institutes of
Health, United States

**Abstract** Voltage-dependent sodium and calcium channels in pain-initiating nociceptor neurons are attractive targets for new analgesics. We made a permanently charged cationic derivative of an N-type calcium channel-inhibitor. Unlike cationic derivatives of local anesthetic sodium channel blockers like QX-314, this cationic compound inhibited N-type calcium channels more effectively with extracellular than intracellular application. Surprisingly, the compound is also a highly effective sodium channel inhibitor when applied extracellularly, producing more potent inhibition than lidocaine or bupivacaine. The charged inhibitor produced potent and long-lasting analgesia in mouse models of incisional wound and inflammatory pain, inhibited release of the neuropeptide calcitonin gene-related peptide (CGRP) from dorsal root ganglion neurons, and reduced inflammation in a mouse model of allergic asthma, which has a strong neurogenic component. The results show that some cationic molecules applied extracellularly can powerfully inhibit both sodium channels and calcium channels, thereby blocking both nociceptor excitability and pro-inflammatory peptide release.

## Introduction

N-type (Cav2.2) calcium channels are present in the presynaptic nerve terminals of many neurons (*Westenbroek et al., 1998*), where they play a major role in synaptic transmission. In particular, N-type channels in primary sensory neurons play a major role in pain/nociceptive signaling (*Park and Luo, 2010*; *Hatakeyama et al., 2001*; *Saegusa et al., 2001*; *Sluka, 1997*) and they are therefore an attractive target for inhibiting pain. In fact, one of the few new treatments for pain introduced in the last 50 years is intrathecal application of ziconotide, a highly potent peptide blocker of N-type calcium channels (*Garber, 2005*). While validating N-type calcium channels as a promising pharmacological target for treating pain, ziconotide has many practical drawbacks, as it is a large peptide with three disulfide bridges that is expensive to synthesize and must be administered intrathecally. Therefore, development of small molecule N-type blockers is an attractive strategy being actively pursued (reviewed by *Lee, 2013*; *Zamponi et al., 2015*; *Zamponi, 2016*; *Patel et al., 2018*; *Yekkirala et al., 2017*).

Compared to the pharmacology of L-type (Cav1) calcium channels, which has been explored for many years, the pharmacology of N-type channels is still at an early stage. A number of L-type

calcium channel inhibitors, including nifedipine, verapamil, and diltiazem, are widely-used clinically for treating arrhythmia and hypertension. The binding sites for the major classes of L-type calcium channel blockers have been defined (reviewed by *Catterall and Swanson, 2015*; *Tang et al., 2016*) and the characteristics of inhibition have been studied in detail. In contrast, while the binding sites for peptide toxin inhibitors of N-type channels have been defined (*Feng et al., 2003*; *Zamponi et al., 2015*; *Bourinet and Zamponi, 2017*), there is no such knowledge for small molecule inhibitors of N-type channels. In fact, most have been described based on screens that provide only minimal characterization of how they interact with channel gating or permeation.

There are general features that appear to be shared by many small-molecule inhibitors of voltage-dependent sodium and calcium channels. The canonical small-molecule inhibitors of voltage-gated channels are local anesthetics like lidocaine, which inhibit voltage-dependent sodium channels by a mechanism that involves tight binding to open and inactivated states of the channel, with relatively weak binding to the closed resting state of the channel (*Hille, 1977*). This property results in channel inhibition that is promoted by steady depolarization (voltage-dependent inhibition) or by repeated cycling through the open state (use-dependence), which occurs if the recovery from depolarization-induced binding is slow enough. Most small-molecule inhibitors of calcium channels also show voltage-dependent inhibition. This behavior is characteristic of inhibition of L-type channels by nifedipine, diltiazem, and verapamil (*Catterall and Swanson, 2015*) and has also been described for many small molecule inhibitors of N-type calcium channels (*Zamponi et al., 2015*; *Zamponi, 2016*).

A particularly interesting feature of local anesthetic inhibition of sodium channels was revealed by studies of permanently charged cationic derivatives of the molecules, notably QX-314 (N-ethyl-lidocaine). QX-314 is ineffective in blocking neuronal sodium channels if applied externally (*Frazier et al., 1970*), apparently because it is too large to enter the outer mouth of the channel, but blocks effectively when applied to the internal (cytoplasmic) face of the channel, presumably because the inner mouth is larger. With intracellular application, QX-314 blocks sodium channels in a highly use-dependent manner because it enters channels only when the channels are open and is trapped inside channels when drug-occupied channels close (*Strichartz, 1973*; *Yeh, 1978*; *Cahalan and Almers, 1979*). The inability of QX-314 to inhibit from the outside of the cell can be exploited to selectively block excitability of nociceptor neurons expressing channels like TRPV1 and TRPA1, whose pores are large enough to pass the QX-314 cation into the cell, effectively silencing only these neurons and producing a selective elimination of pain or itch by blocking action potential conduction to the CNS (*Binshtok et al., 2007*; *Roberson et al., 2011*; *Puopolo et al., 2013*).

Spurred by the relative lack of knowledge about the pharmacology of N-type channel blockers and by the possibility of pursuing a strategy of large-pore channel-mediated drug entry to selectively block transmitter release from nociceptors, we explored the channel blocking properties of a cationic N-type channel inhibitor based on a scaffold described for a group of neutral N-type channel blockers (*Seko et al., 2001*). Surprisingly, we found that the cationic molecule effectively inhibits Cav2.2 channels when administered externally, as well as internally, and actually acts more rapidly with external application. Even more surprisingly, we found that the compound also very effectively inhibits both tetrodotoxin-sensitive and tetrodotoxin-resistant voltage-gated sodium channels with external application, producing use-dependent sodium channel inhibition more potently than lidocaine or bupivacaine. In in vivo tests, the cationic molecule was highly effective in producing long-lasting analgesia in mouse models of surgical and inflammatory pain and was also very effective in reducing lung inflammation in an ovalbumin-sensitization model of allergic airway disease.

## Results

### Voltage-dependent inhibition of Cav2.2 channels by neutral NNCB-2

Our strategy was to design a neutral compound with reasonably potent N-channel blocking activity that could be readily derivatized into a cationic molecule. *Figure 1* (inset) shows the neutral molecule that we designed for this purpose. Its structure is based on a scaffold of a family of molecules with blocking activity against N-type calcium channels described by *Seko et al. (2001)*, although it is not identical to any of the compounds they described. This molecule, which we call NNCB-2 ('NNCB' for 'neutral N-type calcium channel blocker') has a secondary amine group that can be readily converted to a quaternary form to form a cationic molecule, which we call CNCB-2 ('CNCB'

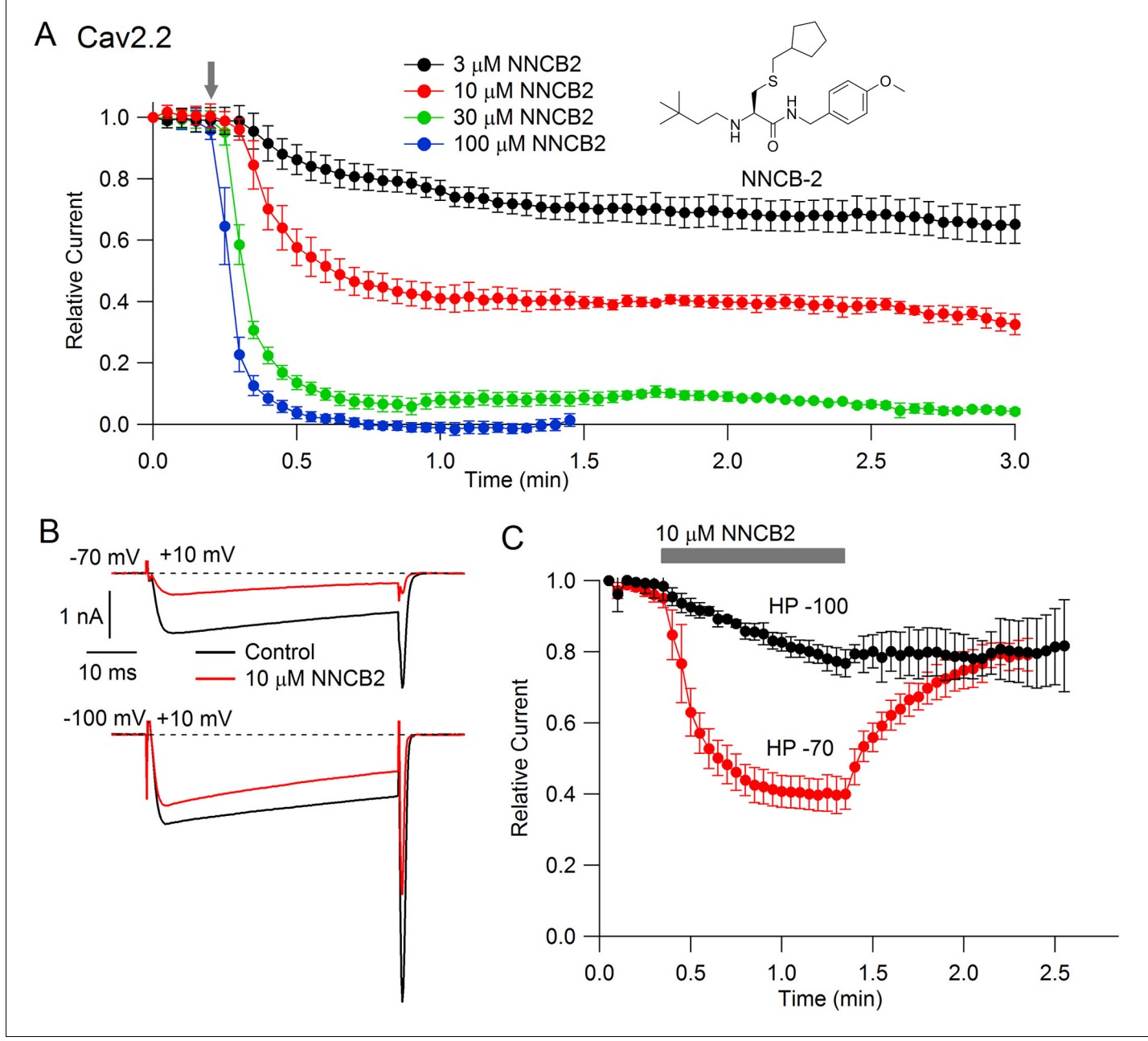

**Figure 1.** State dependent inhibition of Cav2.2 channels by the neutral compound NNCB-2. (**A**) Time course of inhibition of Cav2.2 current by external NNCB-2 applied at concentrations from 3 to 100 µM. Current carried by 5 mM Ba$^{2+}$ was evoked by 50-millisecond steps from −70 to +10 mV delivered every 3 s. Data points show mean SEM (n = 4). Inset: Structure of NNCB-2 (3-Cyclopentylmethylsulfanyl-2-(3,3-dimethyl-butylamino)-N-(4-methoxy-benzyl)-propionamide). (**B**) Inhibition by 10 µM NNCB-2 of Cav2.2 current with current evoked from a holding potential of −70 mV (top) or −100 mV (bottom). (**C**) Test for reversibility of NNCB-2 following exposure for 1 min, using 50 ms test pulses to +10 mV from a holding potential of either −70 mV (n = 3) or −100 mV (n = 7). *Source data 1*.

for 'cationic N-type calcium channel blocker'). We first examined the efficacy with which the neutral compound inhibits cloned N-type calcium channels (*Figure 1*). We found effective and dose-dependent inhibition, with 10 µM NNCB-2 inhibiting calcium channel current by about 60% and 100 µM NNCB-2 producing complete inhibition in about 30 s, when applied at a holding potential of −70 mV, near typical neuronal resting potentials.

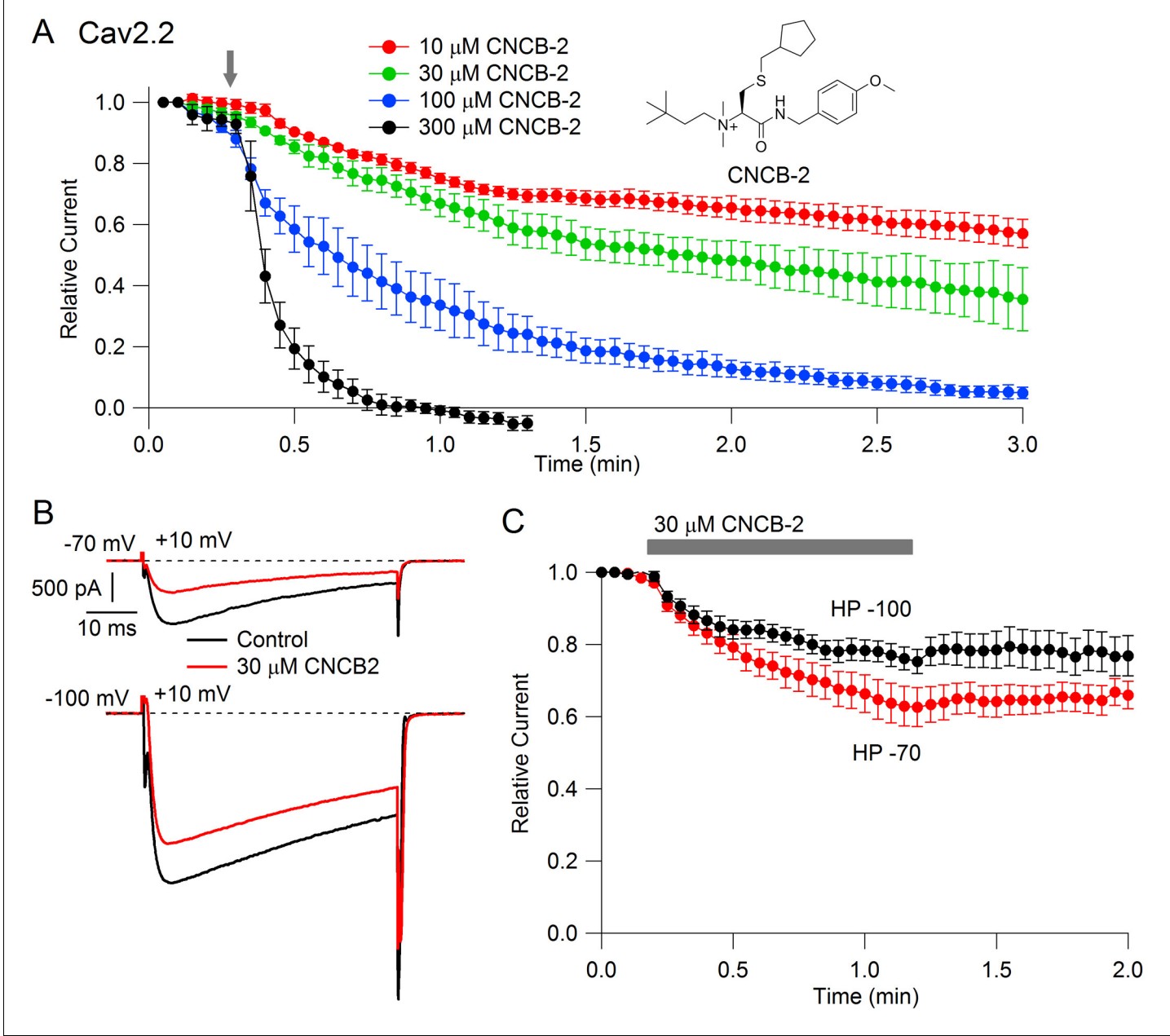

**Figure 2.** State dependent inhibition of Cav2.2 channels by the charged compound CNCB-2. (**A**) Time course of inhibition of Cav2.2 current by external CNCB-2 applied at concentrations from 10 to 300 µM. Current carried by 5 mM $Ba^{2+}$ current was evoked by 50-millisecond steps from −70 to +10 mV delivered every 3 s. Data points show mean ± SEM (n = 4). Inset: Structure of CNCB-2 ([2-Cyclopentylmethylsulfanyl-1-(4-methoxy-benzylcarbamoyl)-ethyl]−(3,3-dimethyl-butyl)-dimethyl-ammonium, chloride salt). (**B**) Inhibition by 30 µM CNCB-2 of Cav2.2 current with current evoked from a holding potential of −70 mV (top) or −100 mV (bottom). (**C**) Minimal reversal of CNCB-2 inhibition by washing following exposure for 1 min, using 50 ms test pulses to +10 mV from a holding potential of either −70 mV (n = 9) or −100 mV (n = 7). Source data can be found in **Source data 1**.

Inhibition by NNCB-2 was voltage-dependent. Compared to inhibition when applied at −70 mV, application at a holding potential of −100 mV produced weaker inhibition that developed more slowly (**Figure 1C**).

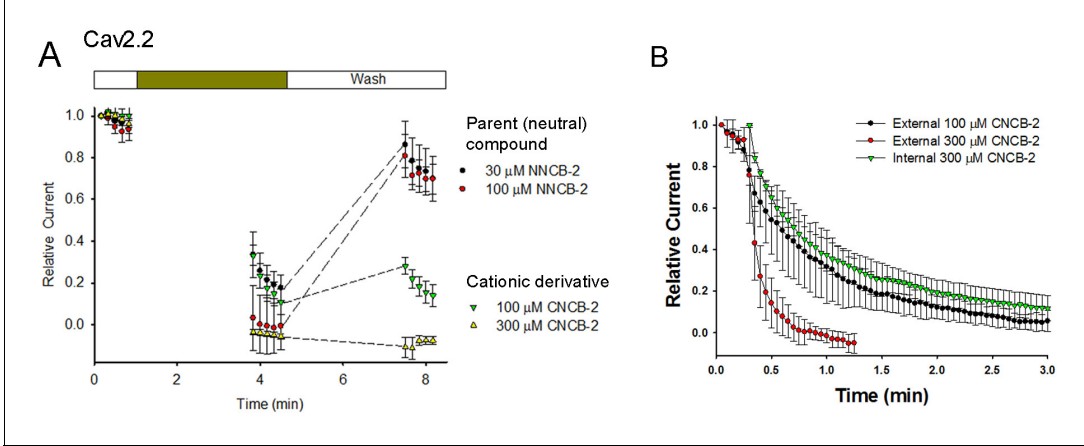

**Figure 3.** Test of whether Cav2.2 inhibition by NNCB-2 and CNCB-2 requires channel opening and comparison of internal and external application of CNCB-2 . (**A**) Calcium channel inhibition by NNCB-2 or CNCB-2 does not require channel opening. $Ba^{2+}$ current was elicited in a whole-cell recording by 50-millisecond steps from −70 to + 10 mV delivered every 10 s. Five pulses were applied in control solution, then NNCB-2 (n = 4) or CNCB-2 (n = 4 to 5) was applied for 3 min with no activation of channels and then depolarizing pulses were resumed for 50 s (in the continuing presence of drug). Then, the cell was perfused with drug-free solution for 3 min without stimulation and then depolarizing pulses were resumed for 50 s to assay recovery. (**B**) Slow inhibition of N-type $Ca^{2+}$ channels by intracellularly applied CNCB-2. 300 µM CNCB-2 applied in the pipette solution (green triangles) produced inhibition that developed relatively slowly over several minutes (green triangles, mean ± SEM, n = 3). The time course of inhibition was similar to that of 100 µM CNCB-2 applied externally (black circles, mean ± SEM, n = 4) and slower than 300 µM CNCB-2 applied externally (red circles, mean ± SEM, n = 4); data points for external application from *Figure 2*. Source data can be found in *Source data 1*.

## Inhibition of Cav2.2 channels with external application of cationic CNCB-2

Studying the cationic molecule CNCB-2, we first tested external application (*Figure 2*). CNCB-2 inhibited Cav2.2 channels effectively with external application. Inhibition by CNCB-2 was slower than with the same concentration of the neutral compound, with 100 µM CNCB-2 requiring about 3 min to produce complete inhibition. Similar to inhibition by NNCB-2, inhibition by CNCB-2 was enhanced when channels were activated from a resting potential of −70 mV compared to a resting potential of −100 mV, consistent with state-dependent binding.

## Inhibition by external CNCB-2 does not require channel opening

One possible interpretation of the slower inhibition by CNCB-2 compared to NNCB-2 would be that the charged compound can only interact with channels when channels are activated, for example if the cationic molecule enters the channel through the open pore. We examined this possibility by testing whether channel opening is necessary for inhibition to develop, by applying the compound in the absence of depolarizing pulses to open the channel (*Figure 3*). We found that both NNCB-2 and CNCB-2 were able to inhibit effectively even in the absence of any channel opening. Application of 100 µM CNCB-2 for 3 min in the absence of stimulation resulted in a decrease of current to 33 ± 5% of control (n = 4, measured during the first depolarizing pulse after the 3 min application). Thus, inhibition by CNCB-2 can occur in the absence of channel opening. However, the degree of inhibition produced by a 3 min application of 100 µM CNCB-2 in the absence of depolarizing pulses was less than the inhibition produced when depolarizing pulses were given during the application (50-millisecond steps from −70 to +10 mV delivered every 3 s, *Figure 2*), where a 3 min application reduced current to 5 ± 2% of control (n = 4). This is consistent with drug binding being accelerated when channels are open or inactivated, but clearly binding can also occur effectively even in the absence of channel opening. We saw similar effects with application of the neutral NNCB-2: a 3 min application of 30 µM NNCB-2 in the absence of stimulation reduced current to 33 ± 11% of control (n = 4), while application in the presence of stimulation reduced current to 5 ± 1% of control of control (n = 3).

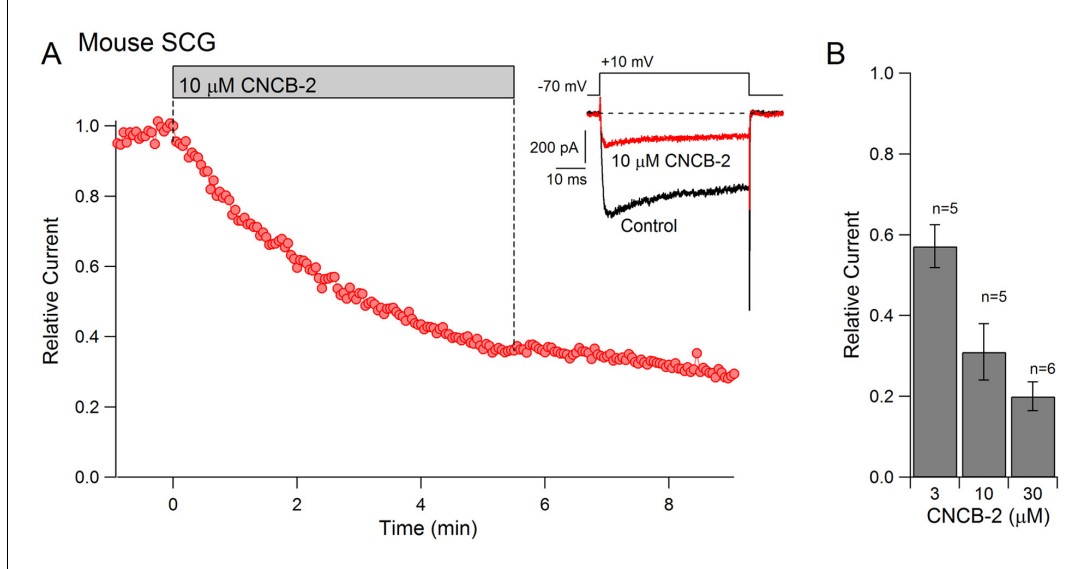

**Figure 4.** Dose-dependent inhibition of native N-type $Ca^{2+}$ channels in mouse sympathetic neurons by CNCB-2 applied extracellularly. (**A**) Time course of inhibition by 10 µM CNCB-2. Current carried by 5 mM $Ba^{2+}$ was evoked by 50 ms steps from −70 to +10 mV delivered every 3 s. The external solution contained 5 µM nimodipine to block L-type calcium channels and 1 µM TTX to block sodium channels. Inset: Current before and after application of CNCB-2. Tail currents in control trace are truncated. (**B**) Current remaining in CNCB-2 applied at 3, 10, or 30 µM (mean ± SEM, n = 5 for 3 and 10 µM CNCB-2, n = 6 for 30 µM CNCB-2). Experiments performed at 37°C. Source data can be found in *Source data 1*.

## Slow recovery from CNCB-2 inhibition

A major difference between the inhibition produced by the cationic and neutral compounds was that recovery from CNCB-2 inhibition was much slower than from NNCB-2 when the agent was removed (*Figure 3A*). With 3 min of wash-out after exposure to 100 µM NNCB-2, current recovered from −1 ± 11% to 81 ± 10% (n = 4) of the initial control value before compound was applied. However, with 3 min of wash-out after exposure to 100 µM CNCB-2, current recovered much less, from 11 ± 6% to 28 ± 4% (n = 4). Thus, although inhibition by CNCB-2 develops somewhat more slowly than that by NNCB-2, CNCB-2 binding to the channel appears to be tighter and reverses more slowly.

## Internal application of CNCB-2

We next tested internal application of CNCB-2. Applying CNCB-2 internally by inclusion in the pipette solution produced inhibition of Cav2.2 calcium channel current that developed over the first several minutes of dialysis (*Figure 3B*). However, inhibition by internal CNCB-2 developed more slowly than with external application. With 300 µM CNCB-2 applied internally, current was reduced to 20% of the initial control value after 114 s, while the same reduction took only 27 s with external application of 300 µM CNCB-2. In fact, the time course of inhibition by internally-applied 300 µM CNCB-2 was similar to the time course of inhibition by 100 µM CNCB-2 applied externally.

## Inhibition of native calcium channels

To examine whether CNCB-2 applied externally can inhibit native N-type calcium channels in neurons as well as cloned N-type channels expressed in HEK cells, we tested its effects on calcium channels in mouse superior cervical ganglion (SCG) sympathetic neurons, a neuronal type in which ~ 80–90% of the overall voltage-dependent calcium current is carried by N-type calcium channels (*Regan et al., 1991*). Externally applied CNCB-2 was effective in inhibiting calcium channel current in SCG neurons, with an $IC_{50}$ near 3 µM (*Figure 4*). As for heterologously expressed channels, inhibition was long-lasting, with very little reversal of inhibition over several minutes of washing with drug-free solution.

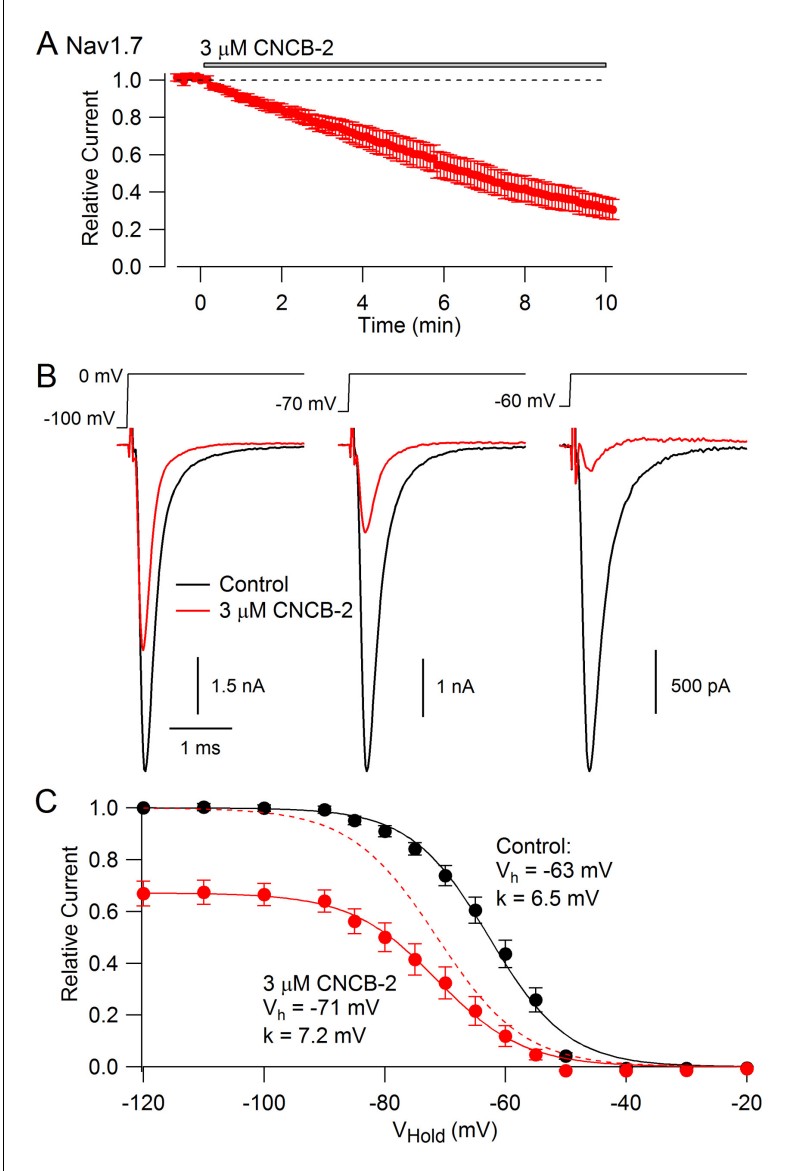

**Figure 5.** CNCB-2 applied extracellularly inhibits Nav1.7 sodium channels. (**A**) Time course of inhibition by 3 μM CNCB-2. Current was evoked by 10 ms steps from −70 to 0 mV delivered every 5 s. Mean ± SEM, n = 5. (**B**) Inhibition by 3 μM CNCB-2 at different holding potentials. Each holding potential was established for 2 s before a test pulse to 0 mV. (**C**) Collected results showing inhibition by 3 μM CNCB-2 as a function of holding potential, with current normalized to largest current in control (from a holding potential of −120 mV). Filled symbols show mean ± SEM, n = 5. Solid lines are drawn according to the Boltzmann equation $1/(1 + \exp((V - V_h)/k))$, where V is holding voltage, $V_h$ is voltage of half-maximal availability, and k is the slope factor. Control: Vh = −63 mV, k = 6.5 mV; 3 μM CNCB-2: Vh = −71 mV, k = 7.2 mV. Dashed red line: fit for CNCB-2 data scaled to 1. Experiments performed at 37°C. Source data can be found in *Source data 1*.

## Inhibition of Nav1.7 channels

We expected that CNCB-2 would have selectivity for N-type calcium channels, because the parent compound was based on a family described as being selective N-type calcium channel inhibitors (*Seko et al., 2001*). Surprisingly, however, we found that CNCB-2 also inhibits sodium channels with external application, acting even more potently on sodium channels than on N-type calcium channels. *Figure 5A* shows the effect of 3 μM CNCB-2 applied to Nav1.7 sodium channels expressed in a stable cell line. With a holding potential of −70 mV and 10-msec depolarizing steps to 0 mV delivered every 5 s, 3 μM CNCB-2 inhibited sodium current to 31 ± 5% of control with application for 10

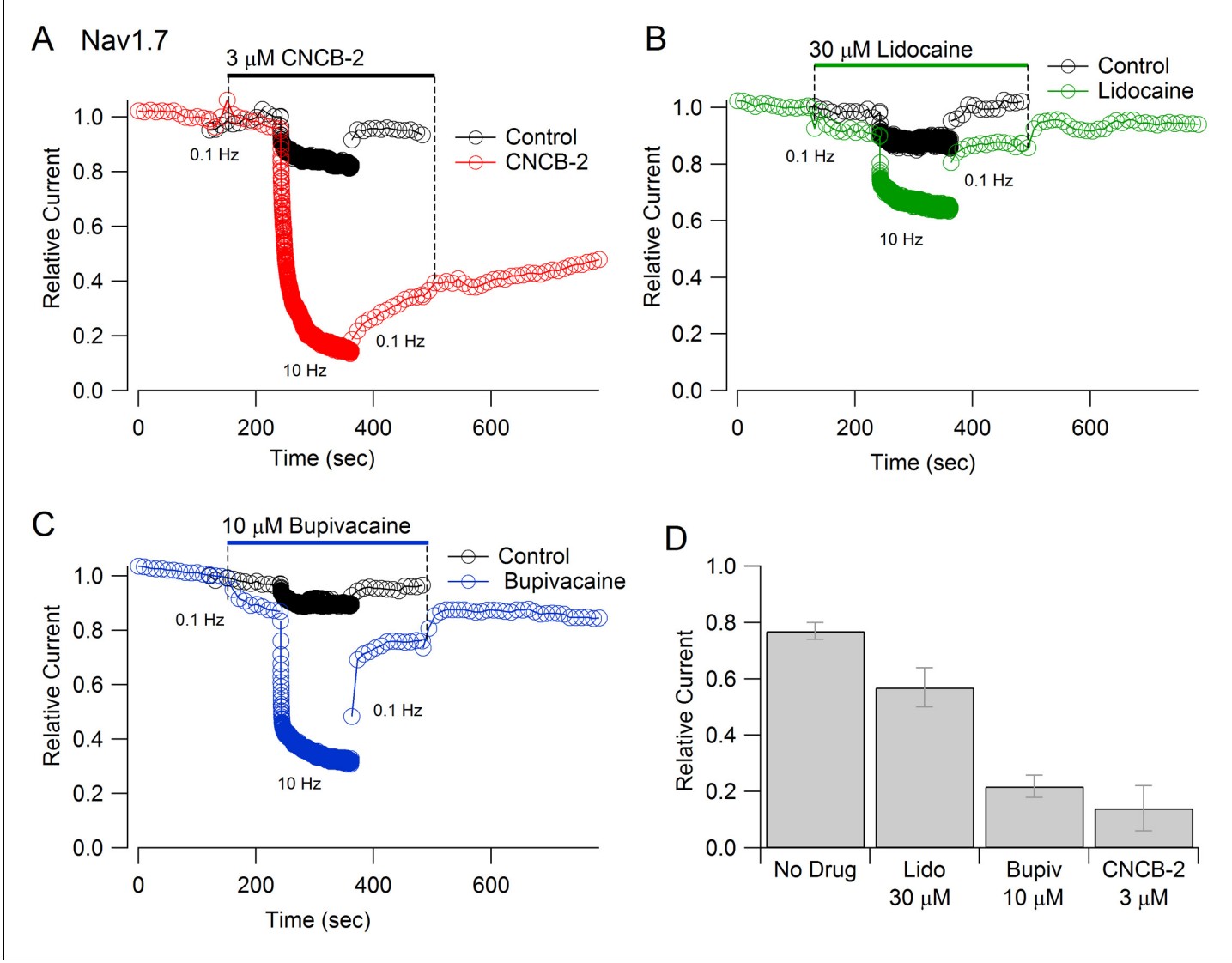

**Figure 6.** Use-dependent inhibition of Nav1.7 sodium channels by externally-applied CNCB-2 compared to lidocaine and bupivacaine. (**A**) Inhibition of Nav1.7 current by 3 µM CNCB-2 applied with stimulation at 0.1 Hz (20-msec depolarizations from −100 mV to 0 mV) for two minutes followed by two minutes of stimulation at 10 Hz, a return to 0.1 Hz stimulation and wash-out of compound. Black symbols: same pulse protocol in the absence of drug, recorded before application of drug. (**B**) Same protocol with 30 µM lidocaine. (**C**) Same protocol with 10 µM bupivacaine. (**D**) Collected results showing current at the end of 2 min of 10 Hz stimulation relative to that before application of drug. Mean ± SEM, n = 12 for no drug, n = 5 for 3 µM CNCB-2, n = 6 for 30 µM lidocaine, n = 5 for 10 µM bupivacaine. Experiments performed at 37°C. Source data can be found in *Source data 1*.

min (n = 5). Inhibition of Nav1.7 channels by CNCB-2 was voltage-dependent, with more potent inhibition from more depolarized holding potentials, in a manner suggesting higher-affinity binding to inactivated states of the sodium channel. Using a protocol in which various holding voltages were established for 2 s in a staircase-like manner, 3 µM CNCB-2 produced equal inhibition at holding voltages of −120 mV and −100 mV (to 67 ± 5% of control at −120 mV and to 67 ± 4% of control at −100 mV), where channels are in the resting state, but much more effective inhibition at more depolarized voltages where more channels are in inactivated states, inhibiting to 12 ± 4% of control from a holding potential of −60 mV where ∼ 55% of channels are in the inactivated state under control conditions (n = 5 for all measurements). The relationship of channel availability to voltage with and without CNCB-2 can be used to estimate the binding affinity of CNCB-2 to resting and inactivated channels, using the relation $\Delta V_h = k*\ln[(1+C/K_R)/(1+C/K_I)]$, where $\Delta V_h$ is the shift in the midpoint of channel availability produced by the compound, k is the slope factor of the Boltzmann availability

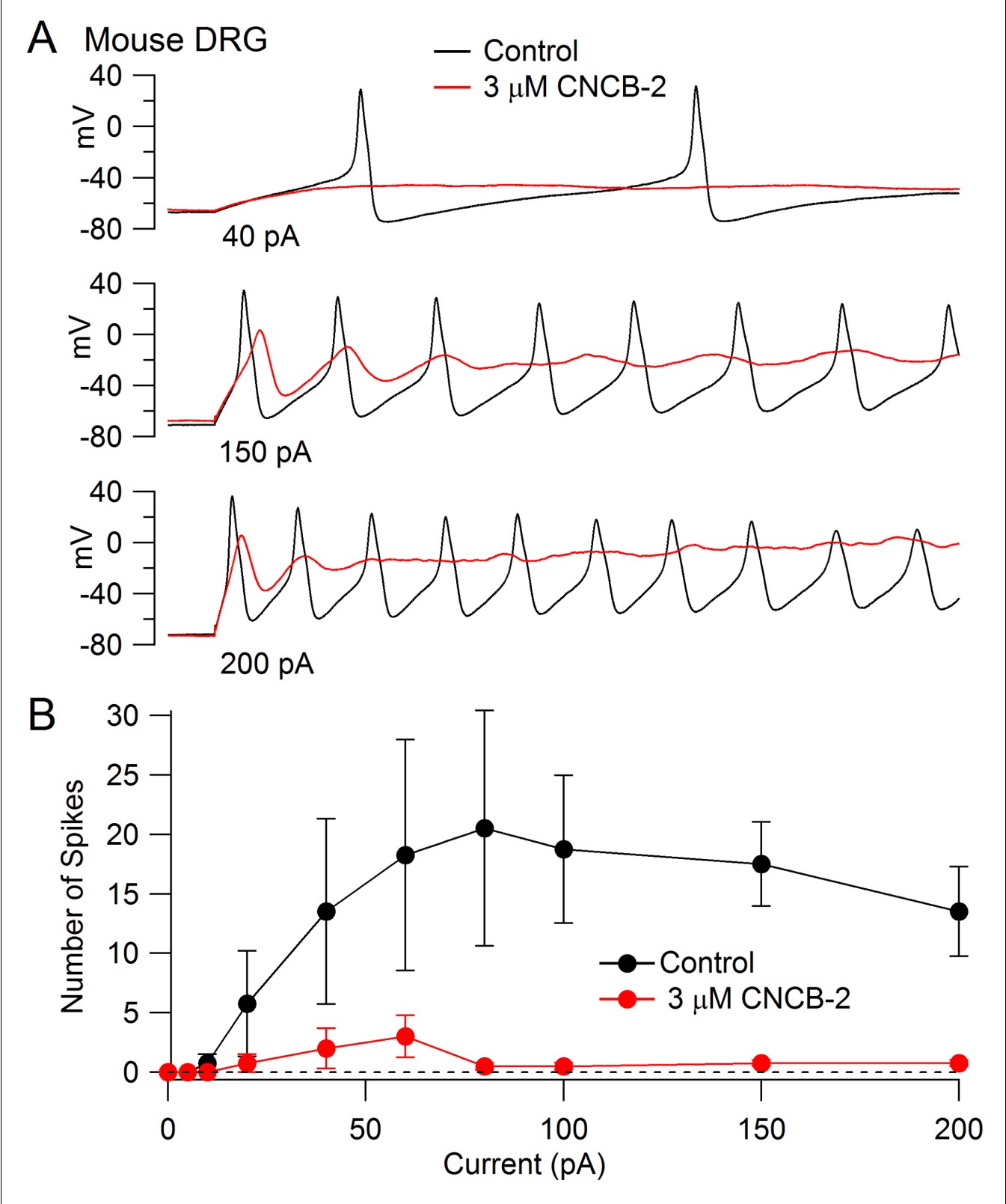

**Figure 7.** Inhibition of action potential generation in mouse DRG neurons by CNCB-2. (**A**) Action potential firing evoked by injections of 40 pA, 150 pA, and 200 pA in a small-diameter mouse DRG neuron before and after 5 min exposure to 3 µM CNCB-2. Resting potential in CNCB-2 was adjusted to match the resting potential in control (near −67 mV) by injection of −20 pA. (**B**) Collected results showing the number of action potentials evoked by 1

*Figure 7 continued on next page*

Figure 7 continued

s current injections of increasing sizes before and after 5 min exposure to 3 µM CNCB-2. Resting potentials in CNCB-2 were adjusted to match the resting potential in control by injection of steady holding current. Mean ± SEM, n = 4. Experiments at 37°C. Source data can be found in *Source data 1*.

curve, C is the concentration of CNCB-2, and $K_R$ and $K_I$ are the equilibrium constants for CNCB-2 binding to the resting and inactivated states, respectively (*Bean et al., 1983*). Making this calculation (and using the inhibition at −120 mV where channels are all in the resting state to estimate $K_R$) results in estimates of $K_R$ = 5.6 µM and $K_I$ = 0.71 µM. This can be compared to the estimates of $K_R$ = 440 µM and $K_I$ = 10 µM for lidocaine binding to sodium channels (*Bean et al., 1983*), implying much tighter binding of CNCB-2 than lidocaine to both resting and inactivated states of the channel.

Inhibition of Nav1.7 channels by CNCB-2 was strongly-use-dependent. Both the rate of inhibition and the degree of inhibition were greater with more rapid stimulation (*Figure 6A*). The use-dependent inhibition by CNCB-2 was more potent than that by lidocaine or even by bupivacaine, the most potent of local anesthetics commonly used clinically (*Becker and Reed, 2012*). In the absence of drug, 2 min of stimulation at 10 Hz reduced current to 77 ± 3% (n = 12), reflecting accumulation into a slow inactivated state from which current recovered quickly on a return to 0.1 Hz stimulation. With 3 µM CNCB-2, current during the 10 Hz stimulation was reduced to 14 ± 8% of control (n = 5), with only slow and partial recovery of current when stimulation was returned to 0.1 Hz. The use-dependent reduction by 3 µM CNCB-2 was more pronounced than by 30 µM lidocaine (reduction to 57 ± 7% of control, n = 6) and similar to 10 µM bupivacaine (reduction to 22 ± 4%, n = 5), with much slower reversal than either lidocaine or bupivacaine when returning to 0.1 Hz stimulation (*Figure 6*).

As for Cav2.2 channels, CNCB-2 applied internally had weaker effects on Nav1.7 channels than when applied externally. In three cells, CNCB-2 applied internally at 3 µM produced very little

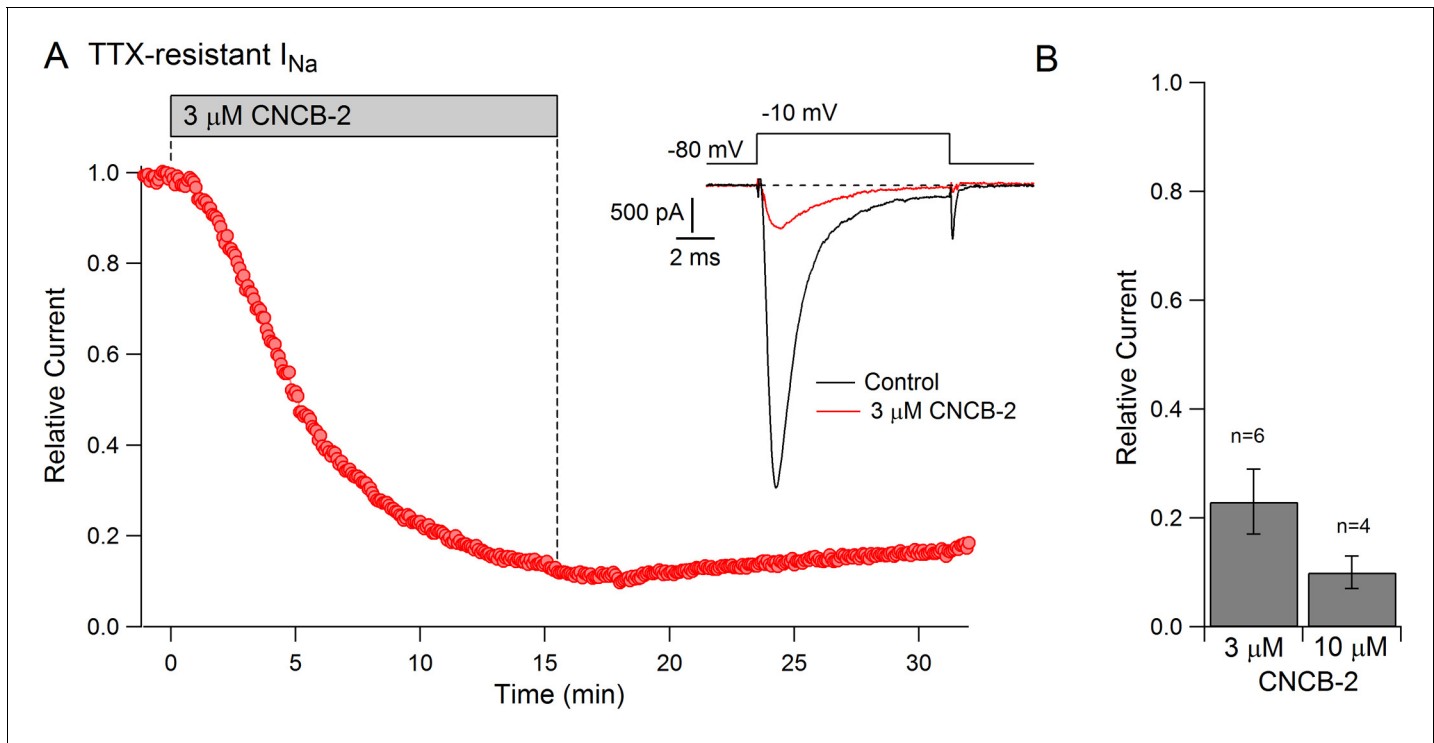

**Figure 8.** Inhibition of TTX-resistant sodium current in mouse DRG neurons by CNCB-2. (**A**) Time-course of inhibition of TTX-resistant sodium current in a small mouse DRG neuron by 3 µM CNCB-2. TTX-resistant sodium current was isolated by solutions containing 300 nM TTX. Current was evoked by a 10-msec step depolarization from −80 mV to −10 mV delivered once every 5 s. Inset: currents before and after CNCB-2. (**B**) Collected results showing current after application of 3 µM CNCB-2 (n = 6) or 10 µM CNCB-2 (n = 4) for long enough to reach steady-state (10–15 min for 3 µM CNCB-2 and 5–10 min for 10 µM CNCB-2). Experiments at 37°C. Source data can be found in *Source data 1*.

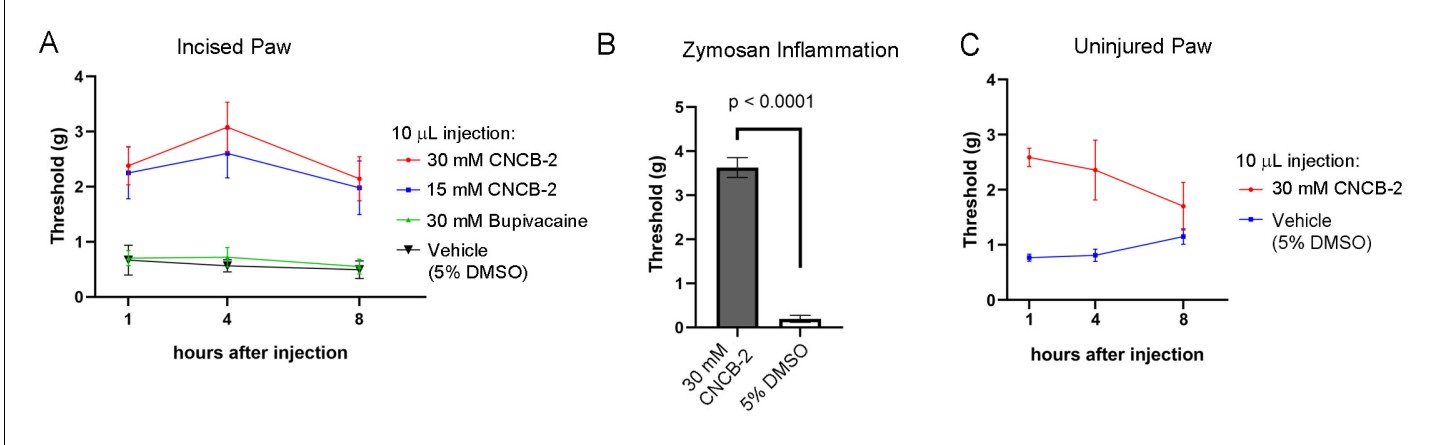

**Figure 9.** Effect of CNCB-2 in mouse pain models. (**A**) Inhibition of paw withdrawal in paw incision model of post-operative pain. Threshold for response to von Frey filaments tested 1, 4 and 8 hr after injection of CNCB-2 into an injured paw 24 hr after the incisional injury (5 mm incision). Mean ± SEM for 50% withdrawal threshold (n = 8 for each group). (**B**) Inhibition by CNCB-2 of paw withdrawal in the zymosan model of inflammatory pain. Either 10 μL of CNCB-2 (30 mM) or vehicle (5% DMSO in normal saline) was injected in hind paw. One hour later the same paw was injected subcutaneously with 20 μL of zymosan (5 mg/ml in saline) and tested 4 hr after the zymosan injection by the von Frey test (Mean ± SEM for 50% withdrawal threshold, n = 5 for each group). (**C**) Direct analgesic effect of CNCB-2: inhibition of paw withdrawal by 10 μL of CNCB-2 (30 mM) injected into uninjured paws (n = 8 for each group). Source data can be found in *Source data 1*.

inhibition even with 2–3 series of 2 min stimulation at 10 Hz. However, when 3 μM CNCB-2 was then applied externally, it immediately produced inhibition during the stimulation at 10 Hz, exactly as when external 3 μM CNCB-2 was applied in the absence of internal CNCB-2. Thus, it appears that CNCB-2 accesses its binding sites on both Nav1.7 and Cav2.2 channels most easily from the external surface of the membrane.

## Inhibition of sensory neuron excitability

The inhibition of Nav1.7 channels suggests that CNCB-2 may reduce electrical excitability of primary sensory neurons, where Nav1.7 channels contribute to overall TTX-sensitive sodium current (*Shields et al., 2012*). We tested this possibility using dissociated dorsal root ganglion neurons, focusing on a population of small diameter DRG neurons that express both TTX-sensitive and TTX-resistant sodium channels, most of which correspond to C-fiber nociceptors. We found that CNCB-2 potently inhibits action potential firing in these neurons. *Figure 7A* shows a typical example, in which the neuron fired multiple large-amplitude action potentials during depolarizing current injections in control conditions but only a single action potential of reduced size after application of 3 μM CNCB-2, even with large current injections. In addition to dramatically depressing action potential firing, application of CNCB-2 produced a depolarization of the resting potential of the neurons, from −76 ± 2 mV in control to −62 ± 3 mV (n = 5) after application of 3 μM CNCB-2. This depolarization of the resting potential was never enough to stimulate firing of the neurons (which had an average threshold voltage of −32 ± 1 mV, n = 8). Because the depression of action potential firing could in principle be partly due to inactivation of sodium channels as a result of the more depolarized resting potential, we also did a series of experiments in which the resting potential of the neurons was adjusted by steady holding current to the same voltage as in control. The action potential firing was also strongly inhibited in these neurons. On average, the number of action potentials evoked by a 1 s 80 pA current injection was reduced from 18 ± 8 in control to 0.6 ± 0.2 after application of 3 μM CNCB-2 (n = 5), similar to the depression of firing in an earlier series in which resting potential was not adjusted (13 ± 2 spikes in control to 2 ± 1 after application of 3 μM CNCB-2, n = 5).

In addition to the reduction in the number of action potentials evoked by current injection, CNCB-2 dramatically depressed the peak of the action potential (from +30 ± 3 mV in control to −2 ± 2 mV in 3 μM CNCB-2, n = 4, holding potential adjusted in CNCB-2).

## Inhibition of TTX-resistant sodium channels

The action potential in small diameter nociceptive DRG neurons is mediated by a combination of TTX-sensitive sodium current, which helps initiate the action potential, and TTX-resistant sodium current, carried mainly by Nav1.8 channels, which generally provides the majority of total sodium current during the action potential (*Renganathan et al., 2001*; *Blair and Bean, 2002*). The dramatic reduction in the magnitude of the action potential by CNCB-2 suggested that CNCB-2 might inhibit TTX-resistant as well as TTX-sensitive sodium channels. We tested this in voltage clamp experiments and found that CNCB-2 effectively inhibited both TTX-sensitive and TTX-resistant components of sodium current in DRG neurons. In a first series of experiments, we found that 30 µM CNCB-2 inhibited overall sodium current in small DRG neurons nearly completely (reduction to 4 ± 1% of control, n = 10, CNCB-2 applied for 2.5–5 min). We then examined TTX-resistant sodium current in isolation and tested lower concentrations of CNCB-2 (*Figure 8*). At 3 µM, CNCB-2 inhibited TTX-resistant sodium current to 23 ± 6% of control (n = 6) and 10 µM CNCB-2 inhibited TTX-resistant sodium current to 10 ± 3% of control (n = 4).

## Effects in a paw incision model of post-operative pain

The efficacy with which CNCB-2 inhibited sodium channels and action potential firing in DRG neurons suggests that it could be an effective analgesic. We tested CNCB-2 in several in vivo rodent models of pain. We first tested it in the rodent paw incision model of post-operative pain (*Brennan et al., 1996*; *Song et al., 2018*; *Cowie and Stucky, 2019*) which produces hyperalgesia mimicking many features of human post-operative pain, including mechanical hypersensitivity, and responds in a similar manner to many pharmacological interventions (*Whiteside et al., 2004*; *Brennan, 2011*). Based on the more potent action of CNCB-2 to produce use-dependent sodium channel inhibition compared to bupivacaine, we tested injections of CNCB-2 at concentrations of 15 mM or 30 mM in a 10 µL volume of injectant and compared their effect with that of 30 mM bupivacaine (~0.5%, the usual concentration of bupivacaine used clinically). Twenty-four hours after the incision, a single injection of 10 µL of either 15 mM or 30 mM CNCB-2 directly adjacent to the incision resulted in a reduction of mechanical sensitivity that lasted for at least 8 hr (*Figure 9A*). In contrast, 30 mM bupivacaine produced no remaining analgesic effect 1 hr after injection, in marked contrast to the long-lasting effects of CNCB-2. The short duration of action of bupivacaine in this model is consistent with a previous study in which bupivacaine injected alone without norepinephrine (generally co-applied with bupivacaine in clinical settings to produce vasoconstriction and thereby slow washout of bupivacaine by blood flow) had effects lasting only 25 min (*Grant et al., 1997*).

## Zymosan model of inflammatory pain

We also tested CNCB-2 in a widely-used mouse model of inflammatory pain, in which intraplantar injection of zymosan into the footpad induces inflammation and mechanical and thermal hypersensitivity, mimicking human inflammatory pain (*Meller and Gebhart, 1997*; *Chiu, 2018*). CNCB-2 was exceptionally effective in this model (*Figure 9B*), increasing the 50% withdrawal threshold from 0.20 ± 0.08 grams to 3.6 ± 0.2 grams (n = 5) of pressure.

## Analgesia of uninjured tissue

The cationic lidocaine derivative QX-314 can inhibit signaling by nociceptors if applied together with activators of TRPV1 or TRPA1 channels (*Binshtok et al., 2007*; *Lennertz et al., 2012*; *Nakagawa and Hiura, 2013*), which have pores large enough to allow entry of QX-314 into neurons. Because CNCB-2 inhibits sodium channels directly from the outside of the cell, there should be no requirement for co-administration with TRPV1 or TRPA1 activators or for endogenous activation of these channels such as might occur during tissue damage or inflammation. We therefore tested the ability of CNCB-2 to inhibit pain signaling when injected into uninjured paws. CNCB-2 produced effective analgesia when injected in paws of naïve mice, with analgesia lasting for at least 4 hr (*Figure 9C*).

## Inhibition of neuropeptide release from cultured DRG neurons

Calcium entry through N-type calcium channels triggers release of neuropeptides from primary sensory neurons by activating vesicular release (*Holz et al., 1988*; *Evans et al., 1996*; *Chi et al., 2009*;

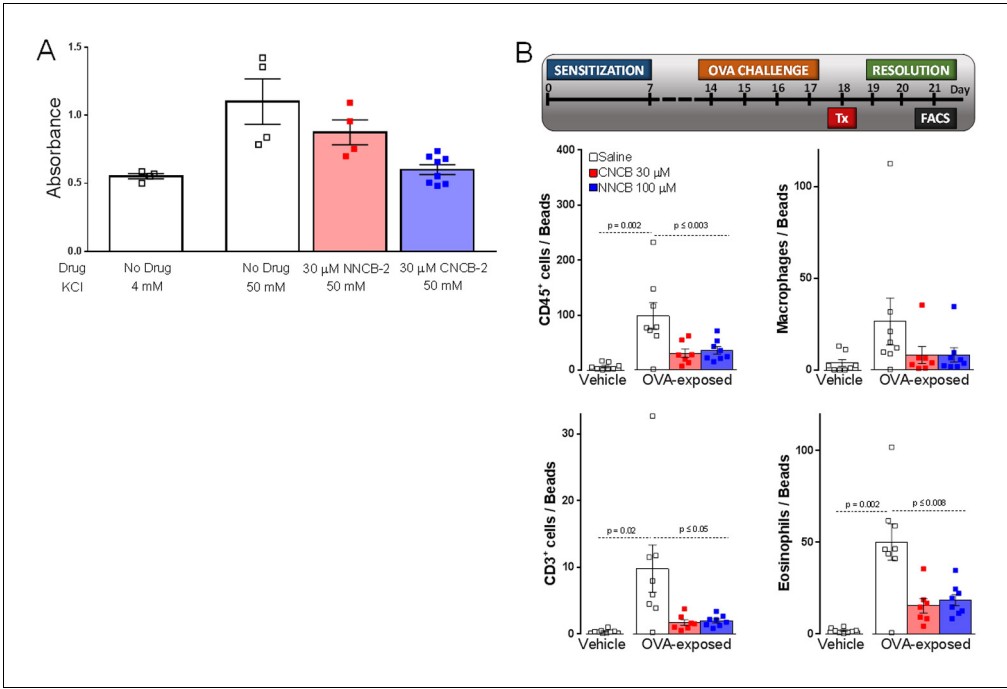

**Figure 10.** Inhibition of CGRP release and neurogenic lung inflammation by NNCB-2 and CNCB-2. (**A**) Effect of 30 μM NNCB-2 or 30 μM CNCB-2 on release of CGRP induced by application of 50 mM KCl to DRG cultures (p=0.15 for 30 μM NNCB-2 (n = 4) compared to 50 mM KCl with no drug (n = 4), p=0.028 for 30 μM CNCB-2 (n = 8) compared to 50 mM KCl with no drug (n = 4); one-tailed t-test for non-homoscedastic data). (**B**) Effect of 100 μM NNCB-2 or 30 μM CNCB-2 on inflammation-associated immune cells in broncheoalveolar lavage fluid in lungs from mice with ovalbumin-induced lung inflammation. p=0.019 for CD45$^+$ cells from ovalbumin-exposed mice application of 100 μM NNCB-2 (n = 8) compared to vehicle (n = 8); p=0.014 for CD45$^+$ cells from ovalbumin-exposed mice with application of 30 μM CNCB-2 (n = 7) compared to vehicle (n = 8). p=0.031 for CD3$^+$ cells from ovalbumin-exposed mice with application of 100 μM NNCB-2 (n = 8) compared to vehicle (n = 8); p=0.028 for CD3$^+$ cells from ovalbumin-exposed mice with application of 30 μM CNCB-2 (n = 7) compared to vehicle (n = 8). p=0.10 for macrophages from ovalbumin-exposed mice with application of 100 μM NNCB-2 (n = 8) compared to vehicle (n = 8); p=0.11 for macrophages from ovalbumin-exposed mice with application of 30 μM CNCB-2 (n = 7) compared to vehicle (n = 8). p=0.008 for eosinophils from ovalbumin-exposed mice with application of 100 μM NNCB-2 (n = 8) compared to vehicle (n = 8); p=0.005 for eosinophils from ovalbumin-exposed mice with application of 30 μM CNCB-2 (n = 7) compared to vehicle (n = 8); one-tailed t-test for non-homoscedastic data. Source data can be found in *Source data 1*.

*Amrutkar et al., 2011*; *Takasusuki and Yaksh, 2011*). We therefore tested the ability of the charged and uncharged inhibitors to reduce release of CGRP, a peptide implicated in pain, migraine and inflammation, from dorsal root ganglion neurons. Studying release of CGRP from cultured DRG neurons induced by depolarization (exposure to 50 mM KCl), we found that both NNCB-2 and CNCB-2 were able to reduce CGRP release, with greater efficacy of inhibition by CNCB-2. CGRP release in this model likely involves calcium entry through L-type (*Evans et al., 1996*) as well as N-type (*Chi et al., 2009*) calcium channels and possibly calcium entry through P-type and R-type calcium channels that are also present in DRG neurons (*Regan et al., 1991*). Further analysis of the calcium entry pathways mediating CGRP release would be required to interpret the greater efficacy of CNCB-2 in this system.

## Action on allergic lung inflammation

Release of pro-inflammatory peptides from primary sensory neurons is part of a neuro-immune positive feedback cycle involved in a number of inflammatory processes with a neurogenic component (*Chiu et al., 2012*; *Foster et al., 2017*), including a mouse model of asthma (*Talbot et al., 2015*). In this model, an allergic reaction to ovalbumin is induced, after which a challenge by nebulized ovalbumin produces airway inflammation, which can be assayed by quantification of immune cells present

in bronchoalveolar lavage fluid. Lung inflammation in this model has a strong neurogenic component and can be greatly reduced either by ablation of primary sensory neurons or by blocking their excitability by QX-314 entry through large-pore channels activated during inflammation (*Talbot et al., 2015*). We tested whether local administration of CNCB-2 could resolve inflammation after it was induced in this model, with the idea that on-going neuropeptide release from the peripheral terminals of sensory neurons may underlie the inflammation. Based on the ability of 30 µM CNCB-2 to reduce peptide release from sensory neurons in vitro, we tested this concentration of CNCB-2 in vivo, applied after lung inflammation had been produced by ovalbumin sensitization and 4 days of daily exposure to nebulized ovalbumin. At this point (day 18 of protocol shown in time-line, *Figure 10B*), lung inflammation is near its peak. With the expectation that reducing neuropeptide release could produce a long-lasting reduction in the neurogenic component of inflammation, the effect of the CNCB-2 exposure was measured 3 days after the CNCB-2 exposure by assaying immune cells in bronchoalveolar lavage fluid. Exposure to 30 µM CNCB-2 dramatically reduced lung inflammation compared to exposure to vehicle, with a reduction of CD3$^+$ cells (lymphocytes) to ~20% of the levels with no CNCB-2 exposure and reductions to 25–35% for the other inflammation-associated immune cell types tested (*Figure 10B*). We compared the effect of 30 µM CNCB-2 with that of 100 µM NNCB-2, selecting a higher concentration based on the lesser efficacy of the neutral compound in the in vitro experiments with CGRP release in *Figure 10A*. In keeping with a greater efficacy of CNCB-2 in this model, the effects of 100 µM NNCB-2 were quantitatively very similar to those of 30 µM CNCB-2. We conclude that topical administration of CNCB-2 is an effective way to inhibit the neurogenic component of allergic inflammation in a long-lasting manner.

## Discussion

Originally, our intention was to design a cationic molecule that would inhibit N-type calcium channels by entering neurons through TRPV1 or other large-pore channels, a strategy we used previously for QX-314 inhibition of sodium channels that can produce long-lasting analgesia (*Binshtok et al., 2007*; *Binshtok et al., 2009*). We had two major surprises in the course of the work: first, that the charged version of the N-type calcium channel inhibitor was more effective in inhibiting N-type calcium channels with external than with internal application, in striking contrast to the charged lidocaine derivative QX-314, which acts only with intracellular application and has no effect when applied extracellularly (*Frazier et al., 1970*; *Binshtok et al., 2007*) and second, that the molecule is also a highly effective inhibitor of sodium channels when applied externally.

For both calcium and sodium channels, inhibition by CNCB-2 was enhanced if applied at more depolarized membrane potentials. This effect most likely reflects a dependence of binding on the gating state of the channel rather than some intrinsically voltage-dependent binding reaction, because the effect saturated when voltages were sufficiently negative to ensure that almost channels are in the resting non-inactivated state. Preferential binding to inactivated states of the channel is characteristic of many non-permanently charged small molecule inhibitors of both sodium and calcium channels (*Lenkey et al., 2010*; *Lenkey et al., 2011*; *Swensen et al., 2012*): *Kuryshev et al., 2014*). Interestingly, however, QX-314 inhibition of sodium channels is qualitatively different from that of uncharged molecules in that QX-314 entry and exit appear to occur only when the channel is actually open (*Cahalan and Almers, 1979*). In contrast, CNCB-2, like lidocaine, can clearly bind to inactivated channels in the absence of any significant channel opening.

Further work will be required to understand the molecular interactions that make it possible for charged CNCB-2 to interact with channels when applied externally. It seems unlikely that the inhibition by externally applied CNCB-2 represents movement through the membrane into the intracellular solution followed by entry into the internal mouth of the pore, because externally applied CNCB-2 acted much more rapidly than the same concentration applied directly inside the cell (for inhibition of both Cav2.2 and Nav1.7 channels). More plausible is access to a binding site on or in the channel that does not require complete translocation of the CNCB-2 molecule across the membrane. One possibility is that there could be partial entry into the bilayer of the more hydrophobic regions of the molecule so that the molecule localizes to the interface between lipid bilayer and the channel, perhaps even the voltage-sensing regions of the channel, analogous to the interaction seen with various toxins that interact with the gating regions of voltage-dependent channels (*Gupta et al., 2015*; *Henriques et al., 2016*). Another possibility is that the CNCB-2 molecule can somehow enter the

pore region of the channel without first translocating to the cytosol. Recent molecular modeling enabled by cryo-EM structures of eukaryotic sodium channels suggests that the charged (protonated) form of lidocaine binds in the pore domain, with the cationic region of the molecule near the selectivity filter, partially displacing sodium ions in the selectivity filter (*Buyan et al., 2018*; *Nguyen et al., 2019*). Modeling shows that the binding site for lidocaine in sodium channels can accommodate two lidocaine molecules (*Nguyen et al., 2019*), suggesting that the same intra-pore binding site could also accommodate the larger CNCB-2 molecule (MW 435 Daltons vs 234 Daltons for lidocaine). An intriguing possibility raised by *Nguyen et al. (2019)* is that a charged molecule like protonated lidocaine could enter the pore through 'fenestrations' in the pore forming regions of Domain III and Domain IV, as the uncharged form of lidocaine likely does (*Gamal El-Din et al., 2018*), and they suggest that external QX-314 might enter the pore of Nav1.5 cardiac sodium channels by this pathway. However, molecular dynamic modeling on the scale of a few μsec cannot directly show the likelihood of such movements, which would be far slower, so this remains speculative. It is also not obvious why in neuronal sodium channels CNCB-2 could enter by this pathway if the smaller QX-314 molecule does not. Further experimental studies with a variety of cationic molecules will be needed to understand the structural characteristics enabling access from the external surface, complemented with the kind of molecular modeling done with lidocaine and QX-314 to understand likely binding sites within the channel. So far such modeling has been done for lidocaine binding to the inactivated state of the channel; similar modeling for interaction with resting state channels would be difficult with lidocaine, since the affinity is so low, but should be more feasible for CNCB-2, with its ~ 100 fold higher affinity than lidocaine for interacting with resting channels.

Molecular modeling may also be useful to understand why inhibition with CNCB-2 applied internally is slower than with external application in both Cav2.2 channels and Nav1.7 channels, a surprising result given the opposite behavior of QX-314. We cannot rule out the possibility that the inhibition by 300 μM internal CNCB-2 seen in Cav2.2 channels reflects slow movement of CNCB-2 molecules from the cytoplasmic side of the membrane to the external leaflet, especially because in these experiments the external medium was not flowing. A more general caveat for interpreting experiments with internal application of compounds in whole-cell recording is the inherent difficulty to accurately control the concentration of the compounds seen at the membrane, which reflects a combination of diffusion from the pipette and entry into or through the bilayer (*Jo and Bean, 2014*). Application with inside-out patches could better define the kinetics and concentration- dependence of internally applied CNCB-2 but such experiments would be technically challenging because of the requirement for long-lived patches given the slow action of CNCB-2.

Sodium channel inhibition by external application has been reported for several other cationic small molecules, notably cationic derivatives of amitriptyline and duloxetine (*Gerner et al., 2003*; *Wang et al., 2016*), tricyclic molecules introduced clinically as antidepressants that have become first-line treatments for neuropathic pain (*Dworkin et al., 2010*) where they may act in part by inhibition of sodium channels (*Dick et al., 2007*). Inhibition by N-methyl-duloxetine was modulated by membrane potential in a manner consistent with preferential binding to inactivated channels, as we saw for CNCB-2 (*Wang et al., 2016*). However, the interaction of CNCB-2 with sodium channels (estimated $K_R$ of 5.6 μM and $K_I$ of 0.71 μM) is far more potent than for N-methyl-duloxetine (estimated $K_R$ of 138 μM and $K_I$ of 2.8 μM), especially for interacting with the resting state of the channel. In fact, the high potency of resting channel inhibition by CNCB-2 stands out when compared with a systematic study of a large range of non-cationic sodium channel inhibitors (*Lenkey et al., 2010*), where the lowest value of $K_R$ found was 21 μM (for the tricyclic antidepressant paroxetine) and most are >100 μM.

This is the first test of a charged inhibitor of N-type calcium channels. A number of previous studies have examined inhibition of cardiac L-type calcium (Cav1.2) channels by charged derivatives of various small molecule inhibitors. Interestingly, different chemical classes of L-type calcium channel inhibitors behave very differently. A charged derivative of the benzothiazepine drug diltiazem behaves much like the QX-314 mediated inhibition of sodium channels, with little effect with external application but able to inhibit with intracellular application in a strongly use-dependent manner, as if the compound can enter channels from the inside and only when channels are open (*Shabbir et al., 2011*). Several charged derivatives of the phenylalkylamine verapamil also produce strong use-dependent inhibition with internal application with no effect on external application (although one derivative did produce significant non-use-dependent inhibition with external application;

*Berjukov et al., 1996*). In contrast, however, charged versions of the dihydropyridine class of calcium channel inhibitors have the opposite behavior, producing inhibition only with external application and having no effect when applied inside the cell (*Kass et al., 1991*). It would be informative to explore possible parallels at the structural level in the ways in which CNCB-2 and charged dihydropyridine molecules are able to interact with various calcium and sodium channels from the outside.

Recent efforts to find new analgesics targeting sodium channels have focused on compounds that have high selectivity for Nav1.7 or Nav1.8 channels (e.g. *Zhang et al., 2010*; *Kornecook et al., 2017*; *Wu et al., 2017*; reviewed by *Yekkirala et al., 2017*) which are expressed in nociceptors but have little or no expression in heart or brain. Selectivity among sodium channel subtypes, or for sodium channels over other channel types, may be important for neutral compounds that are present systemically. However, the ability to non-selectively inhibit a wide range of sodium channels may actually be advantageous for a charged compound administered locally, because nociceptive neurons express a range of sodium channels (*Zheng et al., 2019*) and inhibiting only one of them may not effectively block excitability. This may be a reason for the disappointing clinical results so far (*Yekkirala et al., 2017*), with the new generation of highly selective Nav1.7 inhibitors that bind with high affinity to external sites on Nav1.7 channels by means of a negatively charged 'warhead' region that interacts with externally-oriented parts of voltage-sensing regions (*Ahuja et al., 2015*). Based on its similarly potent inhibition of Cav2.2 channels, Nav1.7 channels, and native TTX-sensitive and TTX-resistant channels, CNCB-2 is likely to inhibit a wide range of voltage-dependent channels. However, systemic exposure is likely to be very low, since the charged molecule is unlikely to penetrate well through tissue into the blood stream, which should minimize unwanted effects in tissues away from the site of application, including the heart and brain, unlike local application of anesthetics like lidocaine and bupivacaine.

It is remarkable that externally applied CNCB-2 inhibits sodium channels more potently than bupivacaine, one of the most potent local anesthetics. Recent work has introduced preparations of bupivacaine delivered in liposomes designed for slow release and longer-lasting effects when applied to wounds during surgery (*Abildgaard et al., 2019*). The intrinsically higher potency and expected slower diffusion away from the site of application – as reflected in the long durations of action we saw in the mouse paw incision model - may make cationic CNCB-2 ideal for such applications even without packaging in liposomes, and its much greater solubility in aqueous solutions may facilitate its use in various other slow-release technologies for even longer-lasting local pain inhibition.

The dual inhibition of both sodium channels and calcium channels may make CNCB-2 or derivatives especially well-suited for treating inflammatory pain. Many kinds of inflammation involve a 'vicious cycle' of interactions between immune cells and nociceptor sensory neuron endings, in which inflammatory mediators from immune cells stimulate the neurons to release neuropeptides like CGRP, substance P, and VIP, which enhance local immune influx by producing vasodilation and increasing capillary permeability and can directly activate immune cells to release more cytokines (*Chiu et al., 2012*; *Foster et al., 2017*; *Liu et al., 2014*; *Luger, 2002*; *Nijhuis et al., 2010*; *Riol-Blanco et al., 2014*; *Talbot et al., 2015*). A key element in neurogenic inflammation is calcium entry into neurons through voltage-dependent calcium channels, which triggers release of peptide-containing vesicles. In particular, N-type (Cav2.2) calcium channels mediate the neuronal release of neuropeptides and are found to be highly expressed in peptidergic primary sensory neurons involved in neurogenic inflammation (*Amrutkar et al., 2011*; *Dunlap et al., 1989*; *Holz et al., 1988*; *Takasusuki and Yaksh, 2011*). Inhibiting N-type channels with CNCB-2 constitutes a novel approach to reducing inflammation by inhibiting release of neuropeptides. Inhibiting N-type calcium channels should inhibit release of all neuropeptides, which may be more effective than targeting only one peptide, as, for example, in the current use of antibodies that target CGRP or its receptor to treat the vascular component of migraine (*Raddant and Russo, 2011*; *Mason and Russo, 2018*). QX-314 also inhibits peptide release, presumably by reducing the depolarization that activates voltage-gated calcium channels (*Talbot et al., 2015*). CNCB-2 direct inhibition of Cav2.2 as well as sodium channels provides two mechanisms of reducing peptide release, which may increase the efficacy of the compound as an anti-inflammatory; CNCB-2 is more effective at 30 μM (*Figure 10*) than 100 μM QX-314 (*Talbot et al., 2015*). Unlike QX-314 which will only enter cells in the presence of activated large pore channels, a feature of active inflammation, CNCB-2 can act even before inflammation is present, potentially preventing its establishment.

We validated this approach to treating inflammation using the clinically relevant mouse model of allergic airway disease (*Foster et al., 2017*) in which inflammation levels can be easily measured by immunophenotyping the number and activation of immune cells in bronchoalveolar lavage fluid. The ability of combined calcium and sodium channel inhibition to reduce allergic airway inflammation is consistent with extensive evidence of a strong involvement of neuropeptide release in this form of inflammation (*Caceres et al., 2009*; *Patterson et al., 2007*; *Talbot et al., 2015*; *Wallrapp et al., 2017*) and with the effect of sodium channel blockade with QX-314 in this model (*Talbot et al., 2015*). Other kinds of disease-associated inflammation known to critically involve release of inflammatory neuropeptides include atopic dermatitis (*Aubdool and Brain, 2011*; *Gouin et al., 2017*; *Misery, 2011*), itch (*Wilson et al., 2013*; *Oetjen et al., 2017*), psoriasis (*Luger, 2002*; *Lotti et al., 2014*), inflammatory bowel disease and colitis (*Chandrasekharan et al., 2008*; *Chandrasekharan et al., 2013*; *Engel et al., 2011*; *Landau et al., 2007*), arthritis (*Krustev et al., 2015*; *Walsh et al., 2015*; *Bersellini Farinotti et al., 2019*) and ocular inflammation (*Mantelli et al., 2010*). We therefore believe this neuronal targeted approach can potentially be used to treat many cutaneous or mucosal inflammatory conditions, where CNCB-2 is well-suited for local application by injection, inhalation, or topical application.

The recent scientific and patent literature has reported a wide range of uncharged compounds effective as N-type calcium channel blockers (*Abbadie et al., 2010*; *Pajouhesh et al., 2012*; *Patel et al., 2015*; *Seko et al., 2002*; *Seko et al., 2003*; *Shao et al., 2012*; *Yamamoto and Takahara, 2009*; *Zamponi et al., 2009*), and many of these have structures that can be modified into cationic versions. Our results suggest that making cationic versions of such molecules that have external blocking activity on calcium and sodium channels may be an unexpected new route to identifying new powerful agents well-suited for both treating pain from injury and surgery and inflammatory conditions with a neurogenic component.

# Materials and methods

## Key resources table

| Reagent type | Designation | Source or reference | Identifiers | Additional information |
|---|---|---|---|---|
| Strain (*Mus musculus*) | Swiss Webster CFW | Charles River | Cat#024 | |
| Strain (*Mus musculus*) | C57BL6/J | Jackson Lab | 000664 | |
| Strain (*Mus musculus*) | BALB/c | Jackson Lab | 001026 | |
| Cell line | rat Cav2.2 tsA201 cell line | PMID: 15201306 | Line #201719 | Dr. Diane Lipscombe, Brown University |
| Cell line | human Nav1.7 HEK cell line | PMID: 22442564 | | Dr. Sooyeon Jo, Harvard Medical School |
| Antibody | FITC anti-mouse CD45 | BD Biosciences | Cat 553079 RRID:AB_394609 | |
| Antibody | PR anti-mouse Siglec-F | BD Biosciences | Cat 552126 RRID:AB_394341 | |
| Antibody | APC anti-mouse GR-1 | Thermo Fisher | Cat 17-5931-81 RRID:AB_469475 | |
| Antibody | PE-Cy7 anti-mouse CD3e | Thermo Fisher | Cat 25-0031-81 RRID:AB_469571 | |
| Antibody | PerCP anti-mouse F4/80 | Biolegend | Cat 123125 RRID:AB_893495 | |
| Commercial assay or kit | MycoAlert PLUS Mycoplasma Detection Kit | Lonza | LT07-703 | |

*Continued on next page*

*Continued*

| Reagent type | Designation | Source or reference | Identifiers | Additional information |
|---|---|---|---|---|
| Commercial assay or kit | Rat CGRP Enzyme Immunoassay Kit | Bertin Pharma/Cayman Chemical | #589001 | |
| Chemical compound | NNCB-2 (3-Cyclopentylmethyl sulfanyl-2-(3,3-dimethyl-butylamino)-N-(4-methoxy-benzyl)-propionamide) | This paper | | Synthesized by Sundia MediTech, structure verified by NMR and mass spectrometry. |
| Chemical compound | CNCB-2 ([2-Cyclopentylmethyl sulfanyl-1-(4-methoxy-benzylcarbamoyl)-ethyl]−(3,3-dimethyl-butyl)-dimethyl-ammonium; chloride salt) | This paper | | Synthesized by Sundia MediTech, structure verified by NMR and mass spectrometry. |
| Chemical compound | Papain | Worthington Biochemical | LS003126 | |
| Chemical compound | Collagenase Type I | Sigma | 10103578001 | |
| Chemical compound | Dispase Type II | Sigma | 4942078001 | |
| Chemical compound | L-15 | Invitrogen | 11415–064 | |
| Chemical compound | Nerve growth factor | Invitrogen | 13290–010 | |
| Chemical compound | Neurobasal A Medium | Invitrogen | 10888–022 | |
| Chemical compound | B-27 | Invitrogen | 17504–010 | |
| Chemical compound | Penicillin-Streptomycin | Invitrogen | 15140–122 | |
| Chemical compound | Fetal calf serum | Invitrogen | 10082–147 | |
| Chemical compound | Minimal Essential Medium | American Tissue Type Collection | EMEM 30–2003 | |
| Chemical compound | Hank's Balanced Salt Solution | Gibco | 14170–112 | |
| Chemical compound | DMEM/F12 | Gibco | 11330–032 | |
| Chemical compound | Tetrodotoxin w/citrate | Abcam | Ab120055 | |
| Chemical compound | Zymosan | Sigma | Z4250 | |
| Software | Clampex | Molecular Devices | Versions 9.2, 10.3.1.5 RRID:SCR_011323 | https://www.molecular devices.com |
| Software | Igor Pro | Wavemetrics | Version 6.12A RRID:SCR_000325 | https://www.wave metrics.com |
| Software | DataAccess | Bruxton Corporation | | http://www.bruxton.com/DataAccess/index.html |
| Software | GraphPad Prism | Graph Pad | RRID:SCR_002798 | Version 5 |

## Cav2.2 recordings

Recordings were made from a cell line stably expressing rat Cav2.2 together with the β3 subunit and the α2δ−1 subunit (line #201719, *Lin et al., 2004*) generously provided by Dr. Diane Lipscombe (Brown University, Providence, RI). Cells were maintained in Dulbecco's modified Eagle's medium

supplemented with 10% fetal bovine serum, penicillin/streptomycin, 25 µg/ml zeocin, 5 µg/ml blasti-cidin, and 25 µg/ml hygromycin (*Lin et al., 2004*) under 5% $CO_2$ at 37˚C. Cell line properties were authenticated by electrophysiology (large voltage-dependent calcium channel currents with kinetic properties expected of this Cav2.2 splice variant, *Lin et al., 2004*) and pharmacology (inhibition by 1 µM ω-conotoxin GVIA). The cell line tested negative for mycoplasma in tests performed every six months (Lonza MycoAlert PLUS Mycoplasma Detection Kit).

Whole-cell recordings were made at room temperature 24–48 hr after plating cells onto cover slips, using patch pipettes made of borosilicate glass with resistances of 3–4 MΩ when filled with the standard internal solution, consisting of 100 mM CsCl, 1 mM $MgCl_2$, 10 mM HEPES, 10 mM BAPTA, 3.6 mM MgATP, 14 mM phosphocreatine, 0.1 mM GTP (lithium salt) and 50 Units/ml creatine phos-phokinase, pH adjusted to 7.2 using CsOH. The external solution consisted of 151 mM TEA-Cl, 5 mM $BaCl_2$, 1 mM $MgCl_2$, 10 mM Glucose, and 10 mM HEPES, pH adjusted to 7.4 with NaOH. A gravity perfusion system connected to a perfusion pencil (AutoMate Scientific) was used to apply compounds to the cell.

Currents were recorded with an Axopatch 200A amplifier, filtered at 5 kHz with a low-pass Bessel filter, and digitized at 10 kHz using a Digidata 1440A data acquisition interface controlled by pClamp10 software (Axon Instruments) and analyzed using Clampfit10, Prizm5, Sigmaplot 10, and Igor Pro 4.0 (Wavemetrics, Lake Oswego, OR). Voltage-activated barium currents were corrected for linear capacitive and leak currents determined using 5 mV hyperpolarizations delivered from the resting potential (usually −70 or −100 mV) and then appropriately scaled and subtracted. Data are given as mean ± SEM.

## Mouse SCG recordings

*Cell preparation.* Dissociated neurons from the mouse SCG were prepared by enzymatic dissocia-tion. SCGs were removed from Swiss Webster (CFW) mice (postnatal day 13–16, either sex), cut in half and treated for 20 min at 37˚C with 20 U/ml papain (Worthington Biochemical, Lakewood, NJ) and 5 mM dl-cysteine in a $Ca^{2+}$, $Mg^{2+}$–free (CMF) Hank's solution (Gibco, Grand Island, NY) contain-ing (in mM): 136.9 NaCl, 5.4 KCl, 0.34 $Na_2HPO_4$, 0.44 $KH_2PO_4$, 5.55 glucose, 5 HEPES, 0.005% phe-nol red, pH 7.4. Ganglia were then treated for 20 min at 37˚C with 3 mg/ml collagenase (Type I, Sigma-Aldrich, St. Louis, MO) and 4 mg/ml dispase II (Boehringer Mannheim, Indianapolis, IN) in CMF Hank's solution. Cells were dispersed by trituration with a fire-polished glass Pasteur pipette in a solution composed of two media combined in a 2:1 ratio: Leibovitz's L-15 medium (Invitrogen, Grand Island, NY) supplemented with 5 mM HEPES, and DMEM/F12 medium (Invitrogen); this solu-tion also had added 100 ng/ml nerve growth factor (NGF) (Invitrogen). Cells were then plated on glass coverslips and were incubated at 37˚C (95% $O_2$, 5% $CO_2$) for 2 hr, after which Neurobasal medium (Invitrogen) containing B-27 supplement (Invitrogen), penicillin and streptomycin (Sigma, St. Louis, MO), and 100 ng/ml NGF was added to the petri dish. Storing the neurons at 4˚C inhibited the growth of neurites, so that cells could be voltage clamped with fast settling of the capacity tran-sient, enabling accurate recording of currents on the fast time scale of the action potential.

Recordings were made using electrodes with resistances of 2–3.5 MOhm when filled with an inter-nal solution containing (in mM): 117 mM CsCl, 4.5 mM $MgCl_2$, 9 mM EGTA, 9 mM HEPES, 14 mM creatine phosphate (tris salt), 4 mM MgATP, and 0.3 mM GTP (tris salt), 9 mM HEPES, pH adjusted to 7.2 with CsOH. Pipette tips were wrapped with thin strips of parafilm to reduce capacitance. Seals were formed in an external solution containing (in mM): 150 mM TEA-Cl, 5 mM $BaCl_2$, 1 mM $MgCl_2$, 10 mM HEPES, and 10 mM Glucose, pH adjusted to 7.4 with TEA-OH. After establishing whole-cell recording, cells were lifted off the bottom of the recording chamber and placed in front of an array of quartz flow pipes (250 µm internal diameter, 350 µm external) using the same external solution with addition of 5 µM nimodipine to block L-type calcium channels and 1 mM TTX to block sodium channels. Experiments were done at 37˚C, using a feedback controller system (Warner Instruments TC-344C) to control the temperature of the flow pipes, which were glued to a rectangular aluminum rod. Voltage clamp recordings were made using a Multiclamp 700B amplifier (Molecular Devices) with currents and voltages controlled and sampled using a Digidata 1322A interface using pClamp 9.2 software (Molecular Devices). Series resistance was compensated by 40–60%. Current and volt-age records were filtered at 5–10 kHz and digitized at 100 kHz. Analysis was performed with Igor Pro (Wavemetrics, Lake Oswego, OR) using DataAccess (Bruxton Software) to import pClamp data. Voltage-activated barium currents were corrected for linear capacitive and leak currents

determined using 5 mV hyperpolarizations delivered from the resting potential (usually −70 or −100 mV) and then appropriately scaled and subtracted. Data are given as mean ± SEM.

## Nav1.7 recordings

Recordings of Nav1.7 currents were made from a cell line stably expressing human Nav1.7 channels previously made in the lab (*Liu et al., 2012*). Cells were grown in Minimum Essential Medium (ATCC) containing 10% fetal bovine serum (Sigma), penicillin/streptomycin (Sigma), and 800 µg/ml G418 (Sigma) under 5% $CO_2$ at 37°C. Cell line properties were authenticated by electrophysiology (large voltage-dependent sodium channel currents) and pharmacology (inhibition by the Nav1.7-selective inhibitor PF-05089771 with previously-reported kinetics and voltage-dependence; *Theile et al., 2016*). The cell line tested negative for mycoplasma in tests performed every six months (Lonza MycoAlert PLUS Mycoplasma Detection Kit). For electrophysiological recording, cells were grown on coverslips for 1 to 6 hr after plating.

Whole-cell recordings were obtained using patch pipettes with resistances of 2–3.5 MΩ when filled with the internal solution, consisting of 61 mM CsF, 61 mM CsCl, 9 mM NaCl, 1.8 mM $MgCl_2$, 9 m M EGTA, 14 mM creatine phosphate (tris salt), 4 mM MgATP, and 0.3 mM GTP (tris salt), 9 mM HEPES, pH adjusted to 7.2 with CsOH. The shank of the electrode was wrapped with Parafilm to reduce capacitance and allow optimal series resistance compensation without oscillation. After establishing whole-cell recording in Tyrode's solution (155 mM NaCl, 3.5 mM KCl, 1.5 mM $CaCl_2$, 1 mM $MgCl_2$, 10 mM HEPES, 10 mM glucose, pH adjusted to 7.4 with NaOH) cells were lifted off the bottom of the recording chamber and placed in front of flow pipes. Recordings were made using a base external solution of Tyrode's solution with 10 mM TEACl. Series resistance was compensated by 40–60%. Recordings were made at 37°C using a Multiclamp 700B amplifier and procedures for recording and analysis followed those for SCG neuron experiments.

## Mouse DRG recordings

*Cell preparation.* Acutely dissociated DRG neurons were prepared as previously described (*Liu et al., 2017*; *Zheng et al., 2019*). Briefly, DRG neurons were removed from Swiss Webster mice (CFW) of either sex (P11-P13), treated for 20 min at 37°C with 20 U/ml papain (Worthington Biochemical, Lakewood, NJ) in calcium- and magnesium-free (CMF) Hank's solution (Gibco, Grand Island, NY) containing 137 mM NaCl, 5.36 mM KCl, 0.33 mM $Na_2HPO_4$, 0.44 mM $KH_2PO_4$, 5.55 mM glucose, 4.17 mM $NaHCO_3$0.02 mM phenol red, pH 7.40 adjusted with NaOH; 300 ~ 310 mOsm. Ganglia were then treated for 20 min at 37°C with 3 mg/ml collagenase (Type I, Sigma-Aldrich, St. Louis, MO) and 4 mg/ml dispase II (Boehringer Mannheim, Indianapolis, IN) in CMF Hank's solution. Cells were dispersed by trituration with a fire-polished glass Pasteur pipette in a solution composed of two media combined in a 1:1 ratio: Leibovitz's L-15 medium (Invitrogen, Grand Island, NY) supplemented with 5 mM HEPES, and DMEM/F12 medium (Invitrogen); this solution also had added 100 ng/ml nerve growth factor (NGF) (Invitrogen). Cells were then plated on glass coverslips and incubated at 37°C (95% $O_2$, 5% $CO_2$) for 2 hr, after which Neurobasal medium (Invitrogen) containing B-27 supplement (Invitrogen), penicillin and streptomycin (Sigma, St. Louis, MO), and 100 ng/ml NGF was added to the petri dish. Cells were stored at 4°C and used within 48 hr.

Whole cell voltage- and current-clamp recordings were made from DRG neurons using electrodes with resistances of 4–8 MOhm, with tips wrapped by Parafilm. Seals were formed in Tyrode's solution (155 mM NaCl, 3.5 mM KCl, 1.5 mM $CaCl_2$, 1 mM $MgCl_2$, 10 mM HEPES, 10 mM glucose, pH 7.4 adjusted with NaOH). After establishing whole-cell recording, cell capacitance was nulled and series resistance was partially (70%) compensated. Cells were lifted and placed in front of flow pipes. For current clamp recording, the internal solution consisted of 140 mM K aspartate, 13.5 mM NaCl, 1.6 mM $MgCl_2$, 0.09 mM EGTA, 9 mM HEPES, 14 mM creatine phosphate (Tris salt), 4 mM MgATP, 0.3 mM Tris-GTP, pH 7.2 adjusted with KOH and an external Tyrode's solution. Reported membrane potentials are corrected for a liquid junction potential of −10 mV between the internal solution and the Tyrode's solution in which current was zeroed before sealing onto the cell, measured using a flowing 3 M KCl reference electrode as described by *Neher (1992)*. For voltage clamp recordings, the external solution was designed to isolate sodium currents by inhibiting calcium channels and potassium channels and contained 155 mM NaCl, 5 mM TEACl, 3.5 mM KCl, 1.5 mM $BaCl_2$, 30 µM

$CdCl_2$, 1 mM $MgCl_2$, 10 mM HEPES, 10 mM glucose, pH 7.4 adjusted with NaOH. Currents through TTX-resistant sodium channels were isolated by including 300 nM TTX in the solution.

Recordings were performed at 37°C using an Axon Instruments Multiclamp 700B Amplifier (Molecular Devices). Procedures for recording and analysis followed those for SCG neuron experiments.

Experiments were done using small diameter DRG neurons (15–30 µm). In voltage clamp experiments, neurons were confirmed to express TRPV1 channels or TTX-resistant sodium currents or both, by testing for currents activated by capsaicin or for sodium currents resistant to 300 nM TTX or for sodium currents with the signature slow inactivation and relatively depolarized steady-state inactivation curves (*Blair and Bean, 2002*). Such tests were not always possible in current clamp experiments but these cells had the wide action potentials with shoulders typical of neurons with TTX-resistant sodium channels.

## Paw incision model

Methods were based on those previously described (*Brennan et al., 1996*; *Song et al., 2018*; *Cowie and Stucky, 2019*). Adult (10–12 weeks old) male C57Bl6/J mice (Jackson Laboratories, Bar Harbor, Maine, USA) were housed in groups of 5 for one week prior to use under controlled conditions (lights on 07:00-19:00 hr; humidity 30–50%; temperature 22–23°C) with food and water available ad libitum. Mice were anesthetized with 3–4% isoflurane and surgery performed under maintenance anesthesia with 1.5–2% isoflurane. Following aseptic preparation with alternating washes of the plantar surface of the paw with 10% povidone-iodine solution and 70% alcohol, a 5 mm longitudinal incision was made on the plantar surface of the right paw with a number 11 surgical blade. The underlying muscles were elevated with forceps to confirm that the muscles and tendons were not damaged by the procedure. The skin was closed with two single sutures of 6–0 nylon. Twenty-four hours after the paw incision, mice were gently restrained in a cloth pocket and received a 10 µL bolus injection of either 15 or 30 mM CNCB-2, 30 mM bupivacaine, or vehicle (5% DMSO) in normal saline to the injured paw adjacent to the incision. Each group consisted of 8 mice, based on previous experiments showing sufficient power to detect significance with 95% confidence. Dosing of compounds was performed by an individual different to the person testing for mechanical hypersensitivity, to ensure blinding. The von Frey testing was performed at 1, 4, and 8 hr after drug injection.

Mechanical hypersensitivity was assessed using the von Frey test as previously described (*Latremoliere et al., 2015*). After animals were habituated to the test cage (7.5 × 7.5×15 cm) with a mesh bottom for 45 min for 2 days, baseline responses were measured using nine von Frey filaments (bending force of 0.04, 0.07, 0.16, 0.4, 0.6, 1, 1.4, 2, and 4 g). Based on the baseline measurement, the mice were assigned to each group so that the baseline mechanical sensitivity among the groups was similar; the test order was randomized. Each filament was applied to the paw three times with 10 s intervals; a withdrawal response was scored if least two of the filament applications elicited flinching or licking of the paw. The up-down method was used to determine the 50% withdrawal threshold following the algorithm of *Gonzalez-Cano et al. (2018)*.

## Intraplantar zymosan injection model

Inflammatory pain was induced by zymosan injection (*Meller and Gebhart, 1997*). Either 10 µL of CNCB-2 (30 mM) or vehicle (5% DMSO in normal saline) was injected subcutaneously in the left hind paw of 8-week-old C57Bl6/J female mice (n = 5 each, based on the paw incision results showing sufficient power to detect significance with 95% confidence). One hour later the same paw was injected subcutaneously with 20 µL of zymosan (5 mg/ml in saline; Sigma, Z4250). Four hours after the zymosan injection, the von Frey test was performed by a blinded investigator.

## CGRP release from DRG neurons

For the CGRP release assay, DRGs were dissected from C57Bl6/J mice into Hank's balanced salt solution, incubated for 90 min at 37°C with 1 mg/ml collagenase A and 2.4 U /ml dispase II (Roche Applied Sciences) in HEPES-buffered saline (Sigma), and then mechanically triturated using glass Pasteur pipettes. DRGs were then centrifuged over a 10% BSA gradient and plated on laminin-coated 96 well plates at approximately 5,000 cells per well. A plate was prepared from each of two

mice, each plate with DRGs in 10 wells. DRGs were cultured 48 hr before conducting a CGRP release assay in B27-supplemented neurobasal-A medium plus 50 ng/ml NGF, 2 ng/ml GDNF, 10 µM arabinocytidine and penicillin/streptomycin. The DRG neurons were exposed to 50 mM KCl with vehicle (2 wells of each plate), 50 mM KCl in the presence of 30 µM NNCB-2 (two wells/plate), or 50 mM KCl in the presence of 30 µM CNCB-2 (four wells/plate), with two wells/plate serving as controls with no KCl depolarization (4 mM KCl), for 10 min at 37°C. The supernatants were collected and analyzed using the Rat CGRP Enzyme Immunoassay Kit (Bertin Pharma/Cayman Chemical, #589001). Plates were read at 405 nm for 0.1 s on a Wallac Victor 1420 Multilabel Counter (Perkin Elmer) and data was analyzed using GraphPad Prism. Data are shown as raw values of absorbance and bars and error bars indicate mean ± SEM.

## Allergic airway inflammation

Allergic airway inflammation was studied in an ovalbumin-based model of type two allergic airway disease (*Haworth et al., 2011*; *Talbot et al., 2015*). On day 0 and day 7, 8-week-old mice (BALB/c, stock number: 001026) were sensitized by a 200 µl i.p. injection of a solution containing 1 mg/ml ovalbumin and 5 mg/ml aluminum hydroxide in phosphate-buffered saline (PBS). On days 14–17 (10:00 am) mice were exposed daily to 6% ovalbumin aerosol for 25 min to produce airway inflammation. On day 18 (near the peak of inflammation in this model), mice were anesthetized using isoflurane and 30 µM CNCB-2 or 100 µM NNCB-2 in PBS was applied intra-nasally as a 50 µL aliquot. On day 21, mice were anesthetized by an intraperitoneal injection of urethane (200 µl i.p., 35%) and a 20G sterile catheter inserted longitudinally into the trachea. To harvest bronchoalveolar lavage fluid, 2 ml of ice cold PBS containing protease inhibitors (Roche) was injected into the lung, harvested and stored on ice. Bronchoalveolar lavage fluid then underwent a 400 g centrifugation (15 min; 4°C), the supernatant was discarded and cells resuspended in 200 µl in FACS buffer (PBS, 2% FCS, EDTA), and incubated with Fc block (0.5 mg/ml, 10 min; BD Biosciences). Cells were then stained with monoclonal antibodies (FITC anti-mouse CD45, BD Biosciences, cat no: 553079, PE anti-mouse Syglec-F, BD Biosciences, cat no: 552126; APC anti-mouse GR-1, eBiosciences, cat no: 17-5931-81; PE-Cy7 anti-mouse CD3e, cat no: 25-0031-81; PerCP anti-mouse F4/80, BioLegend, cat no: 123125; 45 min, 4°C on ice) before data acquisition on a FACS Canto II (BD Biosciences). A leukocyte differential count was determined during flow cytometry analysis of cells expressing the common leukocyte antigen CD45 (BD Pharmigen; cat no: 553079). Specific cell populations were identified as follows: macrophages as F4/80Hi-Ly6gNeg, eosinopohils as F4/80Int-Ly6gLo-SiglecFHi, neutrophils as F4/80Lo-Ly6gHi-SiglecFNeg, and lymphocytes as F4/80Neg-Ly6gNeg-CD3Pos. Total BAL cell counts were performed using a standard hemocytometer, with absolute cell numbers calculated as total BAL cell number multiplied by the percentage of cell subpopulation as determined by FACS (*Haworth et al., 2011*). Data are shown as raw values of cell counts (with outliers included); bars and error bars indicate mean ± SEM.

## Chemicals

NNCB-2 (3-Cyclopentylmethylsulfanyl-2-(3,3-dimethyl-butylamino)-*N*-(4-methoxy-benzyl)-propionamide) and CNCB-2 ([2-Cyclopentylmethylsulfanyl-1-(4-methoxy-benzylcarbamoyl)-ethyl]−(3,3-dimethyl-butyl)-dimethyl-ammonium; chloride salt) were designed based on the scaffold described by *Seko et al. (2001)* and custom synthesized (Sundia MediTech Co, Ltd.). Structures were verified by NMR and liquid chromatography mass spectrometry. An early series of exploratory experiments was done with the iodide salt of CNCB-2 (synthesized by Acesys Pharmatech), which gave similar effects as the chloride salt. For in vitro experiments, compounds were prepared as stock solutions of 30 mM in DMSO; stability of CNCB-2 in the DMSO stock solution was verified by liquid chromatography mass spectrometry. For the in vivo paw pad injection experiments where injection of a small volume (10 µL) was necessary to ensure absorption into the tissue, a stock solution of 600 mM CNCB-2 in DMSO was used to enable use as a final concentration of 30 mM in the 10 µL injectant.

## Acknowledgements

This work was supported by the National Institutes of Health National Institute of Neurological Diseases and Stroke [NS036855, NS110860, NS064274, NS105076 and NS072040], the Defense Advanced Research Projects Agency [award HR0011-19-2-0022], the Blavatnik Biomedical

Accelerator Fund, the Boston Children's Hospital's Technology Development Fund, the Canada Institutes of Health Research (ST), and the Canada Research Chair program (ST). This paper does not reflect the position or the policy of the United States Government, and no official endorsement should be inferred. We are grateful to Dr. Diane Lipscombe for the Cav2.2 stable cell line.

## Additional information

### Competing interests

Seungkyu Lee, Jinbo Lee, Clifford J Woolf, Bruce P Bean: is named as an inventor on a patent application (U.S. Patent Office No. 62/769,420) related to this work. The other authors declare that no competing interests exist.

### Funding

| Funder | Grant reference number | Author |
| --- | --- | --- |
| National Institute of Neurological Disorders and Stroke | NS036855 | Bruce P Bean |
| National Institute of Neurological Disorders and Stroke | NS064274 | Clifford J Woolf |
| National Institute of Neurological Disorders and Stroke | NS072040 | Clifford J Woolf |
| Harvard Medical School | Blavatnik Biomedical Accelerator Fund | Bruce P Bean |
| Boston Children's Hospital | Boston Children's Hospital's Technology Development Fund | Clifford J Woolf |
| Defense Advanced Research Projects Agency | HR0011-19-2-0022 | Clifford J Woolf Bruce P Bean |
| National Institute of Neurological Disorders and Stroke | 110860 | Bruce P Bean |
| National Institute of Neurological Disorders and Stroke | 105076 | Clifford J Woolf |

The funders had no role in study design, data collection and interpretation, or the decision to submit the work for publication.

### Author contributions

Seungkyu Lee, Conceptualization, Data curation, Formal analysis, Supervision, Investigation, Methodology, Writing—original draft, Writing—review and editing; Sooyeon Jo, Sébastien Talbot, Conceptualization, Data curation, Formal analysis, Investigation, Methodology, Writing—original draft, Writing—review and editing; Han-Xiong Bear Zhang, Conceptualization, Formal analysis, Investigation, Methodology, Writing—review and editing; Masakazu Kotoda, Conceptualization, Data curation, Formal analysis, Investigation, Methodology, Writing—review and editing; Nick A Andrews, Conceptualization, Supervision, Investigation, Methodology, Writing—review and editing; Michelino Puopolo, Pin W Liu, Investigation, Writing—review and editing; Thomas Jacquemont, Conceptualization, Data curation, Formal analysis, Investigation, Methodology; Maud Pascal, Data curation, Formal analysis, Investigation; Laurel M Heckman, Formal analysis, Supervision, Validation, Investigation; Aakanksha Jain, Conceptualization, Formal analysis, Investigation; Jinbo Lee, Conceptualization, Resources, Supervision, Writing—review and editing, Designed compounds and oversaw synthesis; Clifford J Woolf, Conceptualization, Resources, Supervision, Funding acquisition, Methodology, Project administration, Writing—review and editing; Bruce P Bean, Conceptualization, Data curation, Supervision, Funding acquisition, Validation, Writing—original draft, Writing—review and editing

## Author ORCIDs

Seungkyu Lee (iD) https://orcid.org/0000-0002-1929-3320
Bruce P Bean (iD) https://orcid.org/0000-0002-5093-3576

## Ethics

Animal experimentation: This study was performed in strict accordance with the recommendations in the Guide for the Care and Use of Laboratory Animals of the National Institutes of Health. All of the animals were handled according to approved Institutional Animal Care and Use Committee (IACUC) protocols of Harvard Medical School ( Protocol 02538) and Boston Children's Hospital (Protocols 19-01-3810 and 19-01-3808).

## Decision letter and Author response

Decision letter https://doi.org/10.7554/eLife.48118.SA1
Author response https://doi.org/10.7554/eLife.48118.SA2

## Additional files

### Supplementary files

- Source data 1. Source data for *Figures 1–10*.

- Transparent reporting form

### Data availability

All data generated or analysed during this study are included in the manuscript and supporting files. Source data 1 contains all data for all figures.

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
