## [Decision Letter]

**Acceptance summary:**

The editors and reviewers enjoyed reading about your surprising discoveries with a charged inhibitor of voltage-activated calcium channels. The results in your beautifully revised manuscript convincingly show that adding a permanent positive charge to a previously described uncharged inhibitor of N-type calcium channels results in a molecule that is surprisingly effective at inhibiting both N-type calcium channels and two types of voltage-activated sodium channels (Nav 1.7 and Nav 1.8) in a state-dependent manner from the extracellular side of the membrane. That your new charged inhibitor is more effective than uncharged local anesthetics for treating both incisional and inflammatory pain suggests that it is an exciting new lead molecule to be developed as an analgesic. Although previous studies with charged local anesthetics suggest that they are only effective at inhibiting sodium channels when applied to intracellular solutions, your surprising discovery of a charged inhibitor that is active on the extracellular side of voltage-activated calcium and sodium channels suggests that more effective new local analgesics might be discovered by adding charges to the growing number of neutral Nav 1.7 inhibitors being developed in search of new analgesics.

**Decision letter after peer review:**

Thank you for submitting your article "A novel cationic N-type calcium channel inhibitor active against neurogenic inflammation" for consideration by *eLife*. Your article has been reviewed by three peer reviewers, including Kenton Jon Swartz as the Reviewing Editor and Reviewer #1, and the evaluation has been overseen by Richard Aldrich as the Senior Editor. The following individual involved in review of your submission has agreed to reveal their identity: Diane Lipscombe (Reviewer #3).

The reviewers have discussed the reviews with one another and the Reviewing Editor has drafted this decision to help you prepare a revised submission.

Summary:

This is an interesting manuscript on the actions of positively charged and neutral small molecule inhibitors of N-type CaV channels. These compounds were synthesized based on previously described amino acid based CaV channel inhibitors, with the goal of extending the groups work on charges local anesthetic blockers of NaV channels that only block from the inside but can enter nociceptors through TRPV1 channels. In this case the charged inhibitor works from both sides, so the authors characterized the concentration and state-dependence of inhibition, showing that they can bind to closed channels but that they inhibit better at depolarized holding voltages. They also found that they are effective inhibitors of CGRP release and neurogenic pain. This is a very nice study, and with great potential to facilitate development of novel analgesics. The following are suggestions for improving the manuscript in revision.

Essential revisions:

1) We feel that it is essential to get more information about whether this compound is selective for Cav2.2 specifically or even calcium channels in general. All the characterization is done on HEK cells stably expressing CaV2.2. What about other channels? What effect does this compound have on the electrical properties of primary sensory neurons? Does it block the endogenous Cav2.2 current? Does it have effects on other ion channels expressed in sensory neurons? What's the IC50 in neurons? Feel free to contact the BRE directly if you need to discuss the scope of any of these suggested experiments, as *eLife* does not wish to ask for very extensive new experiments!

2) The effect of NNCB-2 on high K mediated CGRP release is relatively small and this seems somewhat inconsistent with the effect of NNCB-2 on CaV2.2 currents especially under steady depolarization conditions. Based on Figure 1, 30 μm NNCB-2 should inhibit most CaV2.2 channels. Do the authors have comparable experiments with w-CgTxGIVA or are there reports in the literature to reference? It's possible that other CaV channels are contributing to CGRP release from these neurons.

3) Can the authors provide any information about the mechanism of inhibition or why the charged compound works from both sides of the membrane. Our guess is that these inhibitors are not pore blockers but are interacting with the channel within the membrane. Maybe even the voltage sensors. The ion selectivity filter resides in the external pore in all these channels and would likely be a large barrier to access to the internal pore. It would be great if the authors could provide more mechanistic information, but even if they cannot at this time, it would be good to discuss the potential mechanisms and we would suggest replacing 'blocker' with 'inhibitor' throughout to avoid mechanistic inference.

4) A related point is whether the external solution was flowing during the internal application experiment. If not, perhaps the inhibitor can partition into the membrane and concentrate within the external leaflet where its site of action may reside?

5) How was chemical composition validated? Mass spec? 1D NMR? Some basic information should be provided even in cases where a company was contracted to do the synthesis.

Suggested revisions:

1) The in vivo results while intriguing are only a single assay. What about nocifensive responses (since this is how they introduce their study)? Can it locally injected into the skin (as the authors have done previously with non-permeant sodium channel blockers)? Does the compound attenuate acute heat or mechanical responses? How about under inflammatory or injury conditions?

---

## [Author Response]

Essential revisions:1) We feel that it is essential to get more information about whether this compound is selective for Cav2.2 specifically or even calcium channels in general. All the characterization is done on HEK cells stably expressing CaV2.2. What about other channels? What effect does this compound have on the electrical properties of primary sensory neurons? Does it block the endogenous Cav2.2 current? Does it have effects on other ion channels expressed in sensory neurons? What's the IC50 in neurons? Feel free to contact the BRE directly if you need to discuss the scope of any of these suggested experiments, as eLife does not wish to ask for very extensive new experiments!

To address this point, we began by testing the compound on native N-type calcium channels in mouse sympathetic ganglion neurons. We found that inhibition of native N-type calcium channels is similarly potent as for the cloned channels in HEK cells. This new data is in Figure 4.

In parallel, we also did tests of the compound on excitability of nociceptive primary sensory neurons, as was also suggested. We were very surprised to find that the compound inhibits firing of action potentials very effectively, quite unexpected if it were acting only on calcium channels. This new data is in Figure 7. We also did voltage clamp experiments on sodium channels in nociceptive DRG neurons as well as on Nav1.7 sodium channels expressed in HEK cells, and found that the charged compound inhibits both Nav1.7 and Nav1.8 channels very potently, producing use-dependent inhibition of Nav.7 channels 10 times more potently than lidocaine and three times more potently than bupivacaine, the most potent local anesthetic. This new data is in Figures 5, 6, and 8.

2) The effect of NNCB-2 on high K mediated CGRP release is relatively small and this seems somewhat inconsistent with the effect of NNCB-2 on CaV2.2 currents especially under steady depolarization conditions. Based on Figure 1, 30 μm NNCB-2 should inhibit most CaV2.2 channels. Do the authors have comparable experiments with w-CgTxGIVA or are there reports in the literature to reference? It's possible that other CaV channels are contributing to CGRP release from these neurons.

This is a good point. We have added references to reports in the literature on this point, suggesting that a mixture of calcium channels contribute to peptide release in this system. We have also modified the discussion of these experiments. Because it is likely that the calcium entry triggering CGRP release is through multiple calcium channels, we have removed the implication that the greater efficacy of CNCB-2 in this system is necessarily due to greater efficacy in inhibiting N-type calcium channels. We would have to do many more experiments studying dose-response of both CNCB-2 and NNCB-2 on multiple cloned calcium channels to clarify this. We are mainly interested in the efficacy of CNCB-2, not NNCB-2, as a possible clinical agent, so we have focused on its effects, which are stronger than those of NNCB-2 in both CGRP release and lung inflammation. At this point, the unexpectedly powerful effects of CNCB-2 on sodium channels seemed of more interest to study in detail than a study of effects on other calcium channels, although we plan to do such a study in the future.

3) Can the authors provide any information about the mechanism of inhibition or why the charged compound works from both sides of the membrane. Our guess is that these inhibitors are not pore blockers but are interacting with the channel within the membrane. Maybe even the voltage sensors. The ion selectivity filter resides in the external pore in all these channels and would likely be a large barrier to access to the internal pore. It would be great if the authors could provide more mechanistic information, but even if they cannot at this time, it would be good to discuss the potential mechanisms and we would suggest replacing 'blocker' with 'inhibitor' throughout to avoid mechanistic inference.

This is an important point –thanks. We agree that interaction with the channel within the membrane to affect gating is a more plausible mechanism than pore block, especially given the compound’s ability to inhibit both calcium and sodium channels and its voltage-dependent and use-dependent properties for inhibition of both channels. We have added a discussion of this point and have replaced “blocker” with “inhibitor” throughout.

4) A related point is whether the external solution was flowing during the internal application experiment. If not, perhaps the inhibitor can partition into the membrane and concentrate within the external leaflet where its site of action may reside?

We have added a discussion discussing this possibility, along with the more general caveat about the difficulty of accurately defining the time-course of exposure of the cell membrane to compounds applied in the pipette solution in whole-cell recording, which we have previously explored in comparing lidocaine and QX-314 (Jo and Bean, 2014). Application with inside-out patches would be better for defining the kinetics and concentration-dependence of internally applied CNCB-2 but would be technically very challenging, especially for calcium channels where channel activity runs down quickly in inside-out patches.

5) How was chemical composition validated? Mass spec? 1D NMR? Some basic information should be provided even in cases where a company was contracted to do the synthesis.

Structures were verified by both NMR and liquid chromatography mass spectrometry, with the data furnished by the company who did the synthesis and reviewed by co-author Jinbo Lee, a medicinal chemist. We have added this information to the Materials and methods. We also independently performed liquid chromatography mass spectrometry of stock solutions of CNCB-2 in DMSO to verify stability of the compound in solution (performed by co-author Laurel Heckman, trained as an organic chemist).

Suggested revisions:1) The in vivo results while intriguing are only a single assay. What about nocifensive responses (since this is how they introduce their study)? Can it locally injected into the skin (as the authors have done previously with non-permeant sodium channel blockers)? Does the compound attenuate acute heat or mechanical responses? How about under inflammatory or injury conditions?

This is a great suggestion that became even more important when we found that the compound effectively inhibits sodium channels as well as calcium channels. We originally assumed that the compound was only a calcium channel inhibitor and would only be effective in reducing inflammation associated with neuropeptide release. We choose the mouse asthma model because of previous evidence for a strong involvement of neuropeptides. We have now done a new series of experiments on two different pain models, nocifensive behavior following paw incision and a zymosan-induced inflammation model. We find very effective and long-lasting analgesia in both models, with effects of a single injection lasting at least 8 hours in the paw incision model. CNCB2 is far more long-lasting than bupivacaine in this model. The new data is presented in Figure 9.